# KVCOMM: Online Cross-context KV-cache Communication for Efficient LLM-based Multi-agent Systems

**Hancheng Ye**[1], **Zhengqi Gao**[2], **Mingyuan Ma**[1],
**Qinsi Wang**[1], **Yuzhe Fu**[1], **Ming-Yu Chung**[1], **Yueqian Lin**[1],
**Zhijian Liu**[3], **Jianyi Zhang**[1], **Danyang Zhuo**[1], **Yiran Chen**[1]

[1]Duke University, [2]MIT, [3]NVIDIA
hancheng.ye@duke.edu

## Abstract

Multi-agent large language model (LLM) systems are increasingly adopted for complex language processing tasks that require communication and coordination among agents. However, these systems often suffer substantial overhead from repeated reprocessing of overlapping contexts across agents. In typical pipelines, once an agent receives a message from its predecessor, the full context-including prior turns-must be reprocessed from scratch, leading to inefficient processing. While key-value (KV) caching is an effective solution for avoiding redundant computation in single-agent settings where prefixes remain unchanged, it cannot be directly reused in multi-agent scenarios due to diverging prefixes introduced by agent-specific context extensions. We identify that the core challenge lies in the *offset variance* of KV-caches across agents. To address this, we propose KVCOMM, a training-free framework that enables efficient prefilling in multi-agent inference by reusing KV-caches and aligning cache offsets of overlapping contexts under diverse prefix contexts. KVCOMM estimates and adjusts KV-caches for shared content by referencing a pool of cached examples—termed *anchors*—that store observed cache deviations under varying prefixes. The anchor pool is maintained and updated online, allowing dynamic adaptation to distinct user requests and context structures. KVCOMM achieves over 70% reuse rate across diverse multi-agent workloads, including retrieval-augmented generation, math reasoning, and collaborative coding tasks, all without quality degradation. Particularly, when each fully-connected agent receives 1K input tokens with 512 prefix tokens and 512 output tokens under a five-agent setting, KVCOMM achieves up to $7.8\times$ speedup compared to the standard prefill pipeline, reducing TTFT from $\sim$430ms to $\sim$55ms. Code is available at https://github.com/FastMAS/KVCOMM.

## 1 Introduction

Large Language Models (LLMs) such as GPT-4o [1] and Llama-3 [11] have triggered a surge of interest in collaborative multi-agent systems, where several specialized agents exchange messages to collaboratively solve complex tasks such as retrieval-augmented question answering, mathematical reasoning, and tool-augmented program synthesis [63, 9, 51, 38, 49, 60, 54, 18]. In these settings, every message processed by LLM agents must first go through the prefill stage prior to decoding, during which the model encodes the full conversation history and constructs *key–value (KV) caches*. Although multiple agents often share overlapping context (*e.g.*, retrieved passages or peer outputs), they always redundantly recompute KV-caches for all input tokens, resulting in significant inefficiency of prefilling computation [27, 56, 30], which is defined as a *multi-context redundancy* issue in the

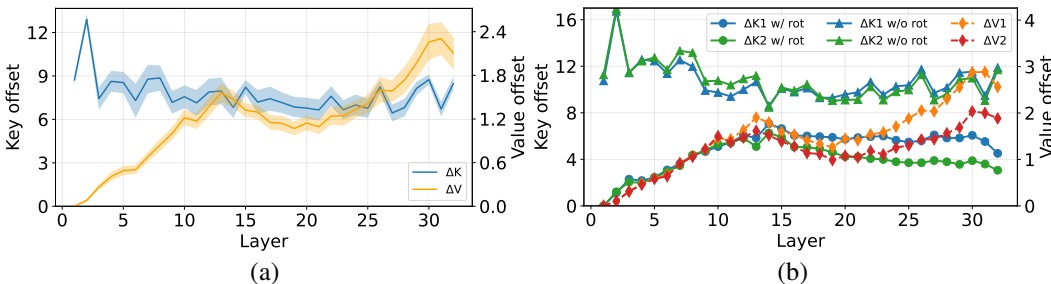

Figure 1: (a) Mean KV-cache offset (measured by $\ell_2$ norm) from the base-context cache for the *same token* across ten distinct prefixes. Shaded regions indicate standard deviation across these prefixes. (b) KV-cache offset comparison between two embedding-similar tokens from the base-context caches when both tokens are prefixed with a new context. "$\Delta K1$ w/ rot" and "$\Delta K2$ w/ rot" represent Key offsets of two tokens with position alignment, respectively. "$\Delta K1$ w/o rot" and "$\Delta K2$ w/o rot" refer to Key offsets without alignment. "$\Delta V1$" and "$\Delta V2$" denote Value offsets of two tokens.

multi-agent system. For example, a single 8B Llama needs ∼430 ms to prefill a 3K-token prompt on one H100 GPU. If each of $M$ agents receives messages from all of its peers, the total prefilling complexity of these repeated computations scales as $\mathcal{O}(M^2)$, posing inefficiency in the utilization of computation and a major challenge for real-time multi-agent collaboration.

Recent works attempt to reduce prefilling overhead primarily through four techniques: prompt-level reuse [10], selective recomputation [58, 27, 56], cache compression [28], kernel-level optimizations [67, 46]. While effective in their target scenarios, these methods share a *fixed* acceleration policy crafted for a particular workload. However, our empirical study reveals that the same shared text can incur vastly different KV deviations once it is preceded by different prefix contexts, *e.g.*, system messages with different roles or upstream agents with different output lengths (see Figure 1a). When the acceleration policy fails to model such an *offset-variance* problem, cache reuse becomes misaligned, causing either large accuracy drops or a fallback to full recomputation that erodes the speed benefits.

This observation motivates a **prompt-adaptive** paradigm that (i) dynamically determines how to reuse KV-caches at *runtime* for each incoming prompt given diverse prefix contexts, and (ii) requires no additional training, profiling, or model modifications, allowing easy adoption on various tasks and agent workloads. To our knowledge, no existing method simultaneously satisfies both desiderata.

In this paper, we introduce *training-free online KV-cache communication* (KVCOMM), a drop-in framework that accelerates multi-agent systems through *shared-context reuse with adaptive KV offsetting*. The key insight is to treat every reuse attempt as an *approximate translation* problem, where the KV-cache of overlapping text becomes reusable for a new prefix once the positional shift and cache offsets from similar samples are identified. As illustrated in Figure 1b, the KV-cache offsets of two similar tokens prefixed with two different prompts present similar distributions across layers, where the deviation of the rotated Key cache is significantly smaller than the unrotated one. Therefore, KVCOMM proposes an **anchor pool** of previously shared samples along with their measured offsets under diverse prefixes. At inference time, the framework first locates the nearest anchor(s) for the requested segment via token similarity (**Anchor Matching**) and then predicts the offset by interpolating their stored deviations, avoiding a replay through the prefilling stage (**Offset Approximation**). For the Key cache update, the cache will first be encoded to the correct position and then biased by the estimated offset, while for the Value cache update, since it has no positional information, the offset is directly added to the cache. Meanwhile, the anchor pool is updated online to catch up with the new input distribution. That is, once the cache of an input segment is predicted as unshareable, it will be marked as an anchor and measure the cache offset under each prefix to extend the reusing range for the subsequent input samples. For the least matched anchors, they would be periodically freed up to save memory consumption and computation cost.

In summary, KVCOMM represents a substantial advancement in adaptively efficient KV-cache sharing in the LLM-based multi-agent system without training or recomputation, offering a practical path toward efficient agent communication. The main contribution is threefold:

- We identify the *multi-context redundancy* as a key challenge for efficient prefilling in the multi-agent scenario, and characterize the *offset-variance problem* that limits traditional KV sharing in such a setting, which to our knowledge, has not been covered by prior work.

- We propose KVCOMM, the first *training-free, prompt-adaptive* cache-sharing framework for efficient prefilling of multi-agent systems, requiring only a few anchors to effectively

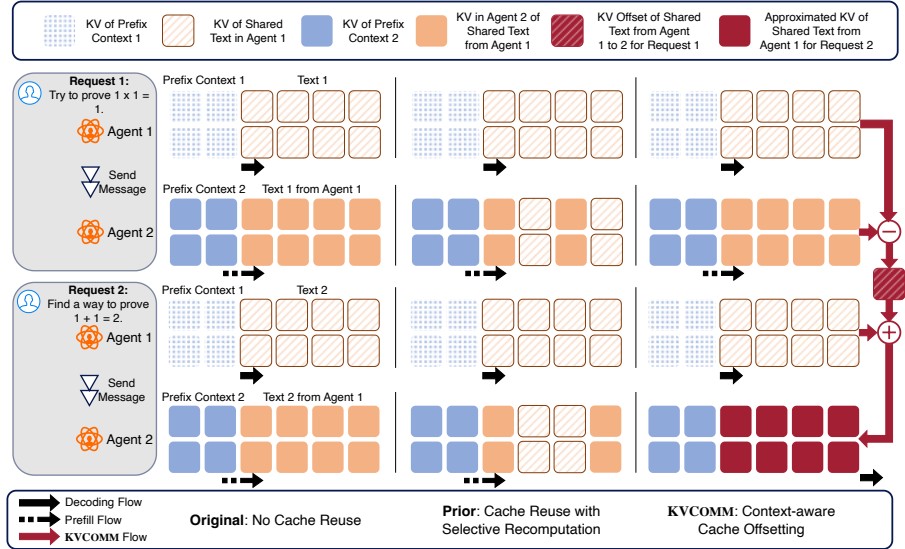

Figure 2: Comparisons with existing KV-cache reuse methods. (Left) The original no-cache-reuse baseline method densely prefills the tokens of all requests. (Middle) Selective recomputation methods [58, 27] select the most critical part of KV-cache for recomputation and reuse the remaining cache of each request. (Right) KVCOMM reuses all KV-caches of the shared context and introduces context-aware cache offsets to align with different prefix contexts, where the context-aware offset refers to the KV-cache deviation induced by the changed prefix context. Such an offset is approximated by the ground-truth ones of previous similar requests. After approximation, the model runner directly starts decoding without prefill.

  approximate KV-cache offsets across different prefix. In KVCOMM, an efficient KV-cache management system is designed to support fast anchor lookup.

- Extensive experiments on three representative tasks with different models, including retrieval augmented generation (RAG), math reasoning, and programming, demonstrate that KVCOMM can achieve $\sim 6.7\times$ average prefill speed-up where each agent is deployed by Llama-3.1-8B-instruct [11] on an NVIDIA H100 GPU. Meanwhile, as the reuse rate reaches 95% across 1,319 samples in a four-agent system for GSM8K [7], KVCOMM achieves comparable performance to the original workload (less than 2.5% accuracy drop).

## 2 Related Work

### 2.1 LLM-Based Multi-Agent Systems

The idea of distributing a complex task across multiple specialized LLM agents has rapidly progressed, from early frameworks such as AutoGPT [35] to mature tool-augmented systems for retrieval, coding, and robotics [17, 26, 39, 63, 9, 48, 22, 50, 51, 62, 55, 3, 31, 52, 24, 15, 42, 19, 40, 23]. Recent studies propose curriculum fine-tuning to promote role specialization [66], graph-structured message routing [57, 69], and hierarchical decision making [32]. Yet practically, each agent still performs a full **prefill** pass for every turn, recomputing the KV tensors over large shared contexts. As agent graphs grow wider or deeper, the prefill complexity of these repeated computations scales quadratically, posing inefficiency in computation utilization. Addressing the prefill bottleneck is thus a prerequisite for scaling multi-agent LLM applications to real-time settings.

### 2.2 KV-cache Acceleration and Reuse

**KV-cache Sharing Scenario.** Prior research has identified three principal patterns for reusing the KV-cache in transformers. (i) *Multi-request sharing* exploits identical prefixes across requests from different users; by copying the KV-cache of the shared prefix, servers can bypass most of the prefill compute when only the tail differs. (ii) *Multi-turn sharing* keeps the cache alive throughout the turns of a single conversation, thus avoiding recomputation of the history. (iii) *Multi-context sharing* handles inputs whose overlapping segment appears at *different contexts*, which aims to filter out the impact of the prefixed prompt in the KV-cache and to combine current context information into the reused KV-cache. Most of these techniques assume that *every agent runs the same model architecture*—typically a vanilla RoPE-based decoder—so that a cached key can be translated by a simple rotation without re-encoding [37, 47]. DroidSpeak [27] extends the sharing from the base model to the fine-tuned one by profiling which layers remain shareable. Current industrial serving

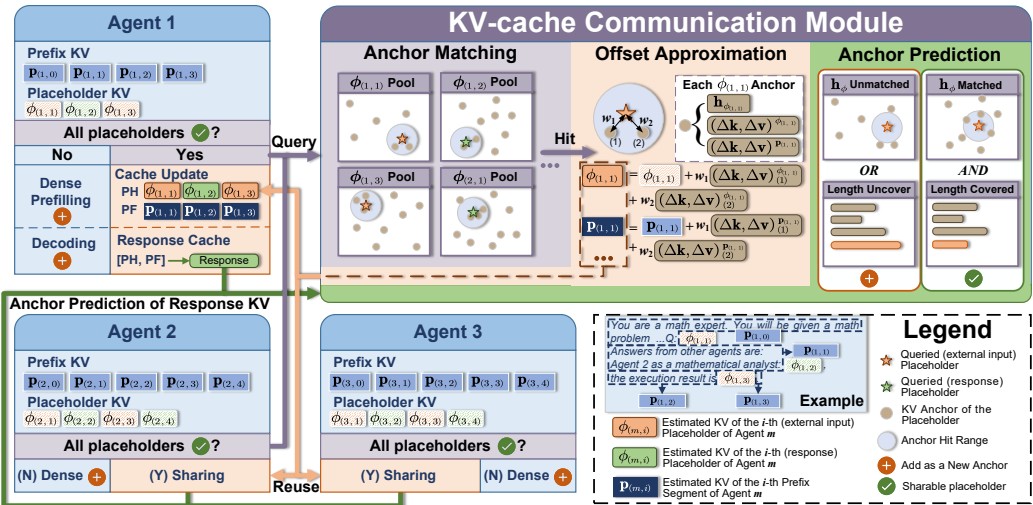

Figure 3: Overview of the KVCOMM framework in a three-agent scenario. Initially, each agent precomputes and stores the KV-cache of prefix segments from its prompt template for future reuse. At runtime, upon receiving a new request, agents check placeholder shareability and query matched anchors. Matched anchors help approximate KV-cache offsets for placeholders and subsequent prefixes through embedding-based interpolation. The matching criteria consider length compatibility and embedding proximity. The updated KV-caches are concatenated for efficient decoding. After decoding, the KV-cache Communication module assesses newly generated caches for potential sharing with other agents based on the established matching rules.

stacks [21, 68, 5] expose the same constraint that architectural identity is a prerequisite for cache reuse.

Existing methods on KV-cache acceleration primarily explore four paradigms. (1) *Prompt-Level Reuse* [10]. PromptCache [10] introduces Prompt Markup Language to explicitly define reusable text segments whose KV-caches are precomputed offline and directly fetched at inference, eliminating recomputation but restricted to fixed prompt structures. (2) *Selective Recomputation* [58, 27, 56]. CacheBlend [58] dynamically identifies and updates tokens exhibiting high variance in KV-caches. DroidSpeak [27] leverages profiling to identify critical attention layers whose KV-caches must be refreshed to maintain accuracy. KVLink [56] further extends it by fine-tuning special tokens and adjusting positional embeddings, enabling KV reuse across multiple document contexts. (3) *Cache Compression* [28]. CacheGen [28] compresses KV-caches into adaptive bit-streams based on available bandwidth; however, the entire token sequence still undergoes compression computations, limiting latency improvements. (4) *Kernel-Level Optimizations* [67, 46]. PrePacking [67] employs a bin-packing strategy to batch variable-length prompts into unified sequences. LoongServe [46] designs Elastic Sequence Parallelism to dynamically manage parallelism strategies and overlap cache migration with decoding steps to enhance GPU utilization. Figure 2 compares the main difference between KVCOMM and existing KV-cache sharing methods. Generally, KVCOMM explores a completely novel paradigm that can reuse all shareable KV-caches regardless of diverse prefix contexts, and align them by leveraging the context-aware cache offsets observed in previous samples.

## 3 Proposed Approach

### 3.1 Preliminaries

**Large Language Models and KV-cache.** Let $\mathbf{x} = [\mathbf{h}^1, \ldots, \mathbf{h}^L]$ denote a list of token embedding sequences, with $\mathbf{h}^l \in \mathbb{R}^{N \times D}$ representing the input token embedding of the $l$-th transformer layer, where $N$ is the number of tokens and $D$ is the feature dimension. For $\mathbf{h}_n^l$ (where $n = 1, 2, \ldots, N$), it is projected by the $l$-th transformer layer to Query, Key, and Value vectors for the subsequent attention computation using: $\mathbf{q}_n^l = R_n W_Q^l \mathbf{h}_n^l$, $\mathbf{k}_n^l = R_n W_K^l \mathbf{h}_n^l$, $\mathbf{v}_n^l = W_V^l \mathbf{h}_n^l$, where $\mathbf{q}_n^l, \mathbf{k}_n^l, \mathbf{v}_n^l \in \mathbb{R}^d$ denotes the $Q, K, V$ values for the $n$-th input token, $W_Q^l, W_K^l, W_V^l$ refer to the corresponding projection weight matrices, and $R_n$ is the position embedding at position $n$, such as rotary position embedding (RoPE) [37]. During autoregressive decoding, the model repeatedly attends to all past positions, and stores every $(\mathbf{k}_n^l, \mathbf{v}_n^l)$ pair in GPU memory, known as *KV-cache*. Therefore, prefilling a prompt of $N$ tokens costs $\mathcal{O}(N^2 d)$ multiply-adds per layer, dominating inference latency for long contexts. Since RoPE applies the fixed rotation matrix to both Key and Query at each position, cached

keys remain valid across subsequent steps with no further arithmetic modification, making KV-cache reuse the primary source of speed-ups in subsequent prefilling and decoding.

**Directed-Graph Multi-agent Systems.** Following [57, 69, 65], we model a multi-agent system as a directed graph $\mathcal{G} = (\mathcal{M}, \mathcal{E})$ whose nodes $m \in \mathcal{M}$ are *agents* and edges $e = (m_s \rightarrow m_t) \in \mathcal{E}$ denote one-way message passing from the $m_s$-th agent to the $m_t$-th agent. At interaction step $t$, the $m$-th agent composes an input prompt $\mathbf{s}_m^{(t)}$ in the template consisting of (i) fixed *prefix segments* shared across all turns, and (ii) *placeholder segments* filled at runtime with user queries, tool results, or upstream agent outputs, as formulated as follows, where $\mathbf{p}_{(m,0)}$ is usually a role-specific system prompt, $\mathbf{p}_{(m,i)}$ is the subsequent prefix segment of the $i$-th placeholder $\phi_{(m,i)}^{(t)}$.

$$\mathbf{s}_m^{(t)} = \left[\, \mathbf{p}_{(m,0)}, \phi_{(m,1)}^{(t)}, \mathbf{p}_{(m,1)}, \phi_{(m,2)}^{(t)}, \mathbf{p}_{(m,2)}, \ldots, \phi_{(m,i)}^{(t)}, \mathbf{p}_{(m,i)} \right]. \tag{1}$$

Our work targets a distinct yet practical setting: a *directed multi-agent graph* in which each node is an **identical RoPE-based LLM checkpoint** instantiated with a role-specific system template. Since agents differ in the length of both prefixes and incoming messages, none of the existing static policies can predict the correct positional and contextual shifts; misalignment either forces full recomputation or yields steep accuracy loss. We therefore develop a *training-free, prompt-adaptive* cache-sharing mechanism, termed KVCOMM, which estimates the true offset on the fly and maintains an online anchor pool to accommodate rapidly changing interaction patterns, reducing prefilling latency without sacrificing task performance. The overall workload of KVCOMM is illustrated in Figure 3, which proceeds as follows.

> **0. Initialization** Before any user requests, all agents precompute and store the KV-caches for all prefix segments defined in their prompt templates.
>
> **1. Placeholder Readiness** When a request arrives, each agent checks whether all placeholders' *base* KV-caches are available. Missing bases are precomputed in parallel. Newly generated placeholder KV-caches are then sent to the *anchor prediction module* to search the anchor pool for similar samples and enable reuse.
>
> **2. Reuse or Fallback** Once all placeholder KV-caches are ready, the agent determines whether reusable KV-cache *deviations* exist for each placeholder. If none are found, standard dense pre-filling is used. For placeholders without reusable deviations, the agent computes the difference between their actual and base KV-caches and stores this deviation in the anchor pool to expand anchor coverage.
>
> **3. Offset Approximation** If all placeholders have reusable deviations, the agent fetches the matched anchors, estimates the KV-cache deviations via Eq. (6) and Eq. (7), and updates the placeholder and prefix KV-caches *in parallel*.
>
> **4. Decoding** The agent concatenates the updated placeholder and prefix KV-caches and initiates response decoding.
>
> **5. Anchor Update** After decoding, the produced KV-cache is passed through the anchor prediction module. If a similar anchor exists, the cache is stored in shared memory for future retrieval. The agent then waits for the next request.
>
> **6. Fallback Storage** If no similar anchor exists, the response KV-cache is stored in the anchor pool so dependent agents can subsequently fill in deviations under their respective contexts; the agent then awaits the next request.
>
> All inter-agent interactions occur through the *KV-cache Communication Module*. When multiple agents share the same user text but use different agent-specific prefixes, we avoid re-running prefilling by treating the KV-cache of the shared text under a new prefix as a *context-dependent offset* from its base KV-cache. We estimate this offset by interpolating from a small set of anchor examples, aligning Key positions via *RoPE de-rotation/re-rotation*, adding the estimated Key/Value offsets, then concatenating the adjusted segments and decoding.

**Positional Alignment is Indispensable.** Before analyzing two arbitrary tokens' cache correlation, we should first solve the position mismatch induced by RoPE. If a token is at position $n$ in one prompt but at $n + \Delta$ in another, the raw keys differ by an orthogonal rotation $R_\Delta$, whose difference can be orders of magnitude larger than the contextual deviation we care about, as demonstrated in Figure 1b. Hence, KVCOMM always de-rotates the stored key by $R_{-\Delta}$ before measuring similarity between

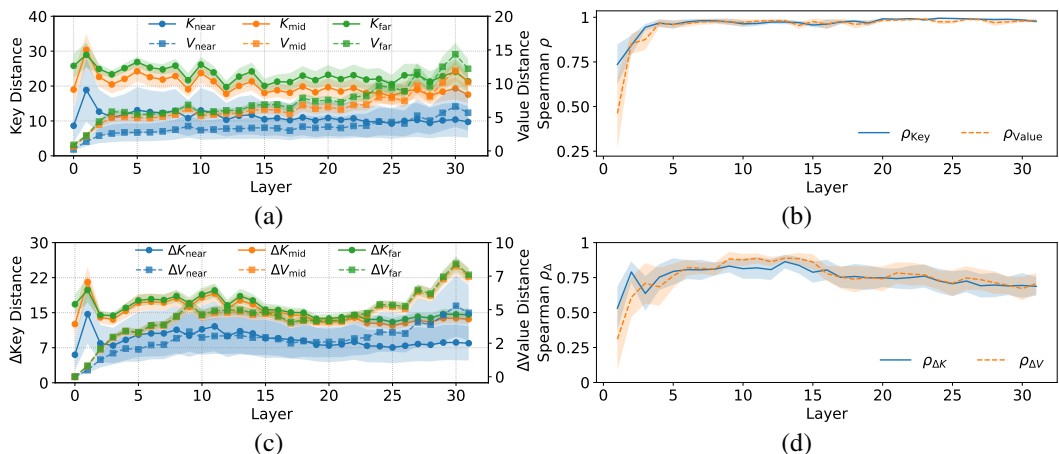

Figure 4: (a)(b) KV-cache proximity experiments for token pairs sharing identical prefixes: (a) shows the KV-cache distances (measured by $\ell_2$ norm) across layers of the token pairs, which are grouped into "near", "mid", and "far" by embedding distance between the two tokens in the token pairs. (b) shows the Spearman [36] correlation between embedding distances and KV-cache proximity across layers. (c)(d) KV-cache offset proximity experiments for token pairs prefixed by two distinct contexts: (c) shows the layer-wise KV-cache offset distances between tokens grouped by embedding proximity. (d) shows the Spearman correlation between embedding distance and KV-cache offset proximity. Experimental details are shown in Appendix 6.3.1.

Key cache offsets of two tokens under the same context change. The following analysis assumes this alignment as completed so that the remaining deviations stem mainly from token identity and context.

## 3.2 Token-level Key/Value Similarity for KV Reuse

**Motivation.** KVCOMM hinges on the empirical observation that per-token KV vectors remain remarkably similar across distinct conversational contexts as long as the model parameters are shared. Intuitively, the residual pathway in every Transformer block keeps a copy of the input representation and adds the attention (Attn) and feed-forward (FFN) refinements:

$$\mathbf{h}_n^{l+1} = \mathbf{h}_n^l + \text{FFN}^l\big(\mathbf{h}_n^l + \text{Attn}^l(\mathbf{h}^l)_n\big), \qquad (2)$$

where $\text{FFN}^l$ and $\text{Attn}^l$ refer to the FFN and Attn modules in the $l$-th layer. Hence, the identity information carried by the original embedding $\mathbf{h}_n^1$ is never overwritten but *accumulates* across layers, suppressing the variation of the projected keys/values. Below we make this insight precise and quantify how far two distinct tokens are when prefixed with the same contexts.

**Proposition 1** (KV-Distance Between Different Tokens). *Let $\mathbf{k}_n^l$ and $\tilde{\mathbf{k}}_n^l$ be the key vectors of two different tokens at position $n$ at layer $l$ that are prefixed with the same token sequence. Assume $\text{Attn}^l$ is $\alpha^l$-Lipschitz, $\text{FFN}^l$ is $\beta^l$-Lipschitz [20]. Define $\sigma^l \triangleq \beta^l(1 + n\alpha^l)$. Then*

$$\big\|\mathbf{k}_n^l - \tilde{\mathbf{k}}_n^l\big\| \leq C_R C_K^l \prod_{j=1}^{l-1}\big(1 + \sigma^j\big)\,\delta_n, \qquad \delta_n = \max_{k \leq n}\big\|\mathbf{h}_k^1 - \tilde{\mathbf{h}}_k^1\big\|, \qquad (3)$$

*where $C_R > 0$ is related to RoPE and $C_K^l > 0$ is related to $l$-th layer projection key matrix $W_K^l$. Similarly, the inequality also holds for the value caches of the two tokens.*

The proof is deferred to Appendix 6.2.2. It can be observed that Eq. (3) bounds the KV distance by the embedding gap scaled through layers, so tokens that start closer in embedding space have tighter bounds and greater cache-reuse potential. Figure 4a and 4b empirically demonstrate this insight, where the KV-caches of "near" token pairs are consistently closer to each other than the other two groups, and the KV-cache proximity is highly correlated to the token embedding distance.

We now examine this relation in multi-agent settings, where two similar tokens face different prefixes.

**Proposition 2** (Deviation Proximity With Different Prefixes). *Let $\mathbf{k}_{n_a}^l$ and $\tilde{\mathbf{k}}_{n_a}^l$ be the key vectors of two different tokens at position $n_a$ at layer $l$ that are prefixed with prompt $\mathbf{p}_a$. Similarly, $\bar{\mathbf{k}}_{n_b}^l$ and $\tilde{\bar{\mathbf{k}}}_{n_b}^l$ are the key vectors of the two tokens at position $n_b$ at layer $l$ that are prefixed with prompt $\mathbf{p}_b$. We denote the key cache deviation of each token at layer $l$ by $\Delta^l = \bar{\mathbf{k}}_{n_b}^l - \mathbf{k}_{n_a}^l$, $\tilde{\Delta}^l = \tilde{\bar{\mathbf{k}}}_{n_b}^l - \tilde{\mathbf{k}}_{n_a}^l$. Under the same Lipschitz assumptions as Proposition 1 and after positional alignment,*

$$\big\|\Delta^l - \tilde{\Delta}^l\big\| \leq 2\,C_R C_K^l \prod_{j=1}^{l-1}(1 + \sigma^j)\delta_{n_a}, \qquad \delta_{n_a} = \max_{k \leq n_a}\big\|\mathbf{h}_k^1 - \tilde{\mathbf{h}}_k^1\big\|. \qquad (4)$$

*Similarly, the inequality also holds for the value caches of the two tokens.*

Proof is in Appendix 6.2.2. Eq. (4) shows that tighter embedding gaps again yield smaller deviation bounds, supporting cross-context offset reuse. Figs. 4c and 4d validate this under the same setup as Figure 4a, with each token evaluated under two distinct prefixes, where the KV-cache offsets of "near" token pairs are also consistently closer to each other than the other two groups, and the KV-cache offset proximity is also highly correlated to the token embedding distance.

The above propositions motivate an anchor-based KV-sharing scheme that stores representative offsets as reusable anchors for future agent interactions to reduce all redundant prefilling latency of agents.

## 3.3 Anchor-based KV-cache Communication

We now introduce our anchor-based communication framework to unify the KV-cache sharing mechanism in multi-agent systems. During setup, each agent extracts placeholder information from its role-specific prompt into a structured dictionary, indicating token positions. The naming conventions for placeholders are detailed in Appendix 6.2.3. Each placeholder initializes an individual anchor pool upon receiving the first sample.

At runtime, subsequent input samples trigger a reuse check across agent placeholders. Agents reuse KV-caches directly from corresponding anchor pools if reuse conditions are satisfied, significantly speeding up inference by skipping redundant prefilling. Otherwise, agents revert to standard prefilling, updating anchor pools with newly computed KV-caches to enrich future reuse opportunities.

**Anchor Pool Design.** An anchor pool stores key information for each placeholder sample: (1) the base KV-cache, computed independently without external contexts; (2) offsets between the base and actual KV-caches within each agent's context; and (3) offsets of subsequent neighboring prefix segment's KV-cache. Thus, an anchor is represented as {ph_name: base KV, agent_id_ph: placeholder offset, agent_id_pf: prefix offset}. Neighboring prefix offsets are crucial due to position-dependent KV-cache shifts introduced by the placeholder's context changes, as highlighted by the sink attention mechanism [53], which emphasizes local contextual dependencies.

**Anchor Prediction.** Determining whether newly-generated KV-caches, *e.g.*, responses, user inputs, etc., could be shared or treated as new anchors involves evaluating the embedding-based proximity [61, 29, 64, 59] and token length compatibility with existing anchors. The prediction criterion is designed as follows:

$$\mathcal{P}_{anchor}(\phi) = (\mathcal{L}_\phi > \max_{\psi \in \mathcal{A}} \mathcal{L}_\psi) \cup (\mathcal{H}_{\phi|\mathcal{A}} > \gamma \log |\mathcal{A}_\phi|), \quad \mathcal{H}_{\phi|\mathcal{A}} = \sum_{\psi \in \mathcal{A}_\phi} w_{\phi \to \psi} \log w_{\phi \to \psi}, \quad (5)$$

where $\mathcal{A}$ refers to the anchor pool that the placeholder $\phi$ belongs to, $\psi$ denotes an anchor in $\mathcal{A}$, $\mathcal{L}_\star$ represents the sequence length of the sample $\star$, $\mathcal{H}_{\phi|\mathcal{A}}$ measures entropy of the embedding-distance-based weights among longer anchors in the anchor pool, $w_{\phi \to \psi} = \texttt{softmax}(-\|\mathbf{h}_\phi - \mathbf{h}_\psi\|), \psi \in \mathcal{A}_\phi$, $|\mathcal{A}_\phi|$ refers to the number of anchors longer than $\phi$ in $|\mathcal{A}|$, and $\gamma$ is a threshold to determine how far a shareable sample could be away from the anchors of $\mathcal{A}$ in the embedding space. Intuitively, anchors closer in embedding space yield more reliable offset predictions (validated by Prop. 2 and Figure 4c), and length compatibility ensures correct positional alignment (see Figure 1b).

**Anchor Update.** When KV-cache sharing criteria are unmet, the newly-generated cache becomes a new anchor's base KV-cache. Agents relying on this unshareable placeholder revert to regular prefilling, providing agent-specific offsets for both the placeholder and its neighboring prefix segments to populate the new anchor entry. Due to GPU memory constraints, we implement an adaptive anchor pruning strategy: once anchor pools reach a predefined size $\mathcal{V}$, the least frequently accessed anchor among the earliest-added entries is discarded, maintaining a relevant and efficient anchor repository.

## 3.4 Anchor-based Cache Update

When placeholders in an agent's prompt are predicted shareable, we efficiently update their KV-caches via anchor matching and offset approximation.

**Anchor Matching.** We retrieve reliable anchors identified during prediction, performing parallel reads due to independent addressing, leading to negligible overhead compared to traditional prefilling.

**Offset Approximation.** Using matched anchors, we approximate placeholders' KV-caches within agent-specific contexts. Neighboring prefix segments are updated similarly based on the placeholder

Table 1: Performance of three cache-management strategies under different numbers of collaborating agents. Accuracy is reported for MMLU and GSM8K (Llama-3.1-8B-Instruct); Pass@1 is reported for HumanEval (Qwen-2.5-coder-7B). Higher is better. In addition, the Reuse Rate is reported for both KVCOMM and CacheBlend. Note that the Reuse Rate for CacheBlend is defined as the proportion of tokens reusing KV-caches in whole token sequences, while the Reuse Rate of KVCOMM is defined as *the frequency of agents reusing all KV-caches in the whole serving procedure*.

| Dataset | Metric | Method | # Agents | | | |
|---|---|---|---|---|---|---|
| | | | 2 | 3 | 4 | 5 |
| MMLU | Accuracy (%) | Original | 47.1 | 66.7 | 68.0 | 69.9 |
| | | CacheBlend | 65.4 | 65.4 | 65.4 | 67.3 |
| | | **KVCOMM** | **64.7** | **68.6** | **68.0** | **69.9** |
| | Reuse Rate (%) | Original | 0 | 0 | 0 | 0 |
| | | CacheBlend | 80 | 80 | 80 | 80 |
| | | **KVCOMM** | **74.5** | **69.9** | **70.1** | **67.6** |
| GSM8K | Accuracy (%) | Original | 81.1 | 82.4 | 82.1 | 81.7 |
| | | CacheBlend | 82.0 | 75.1 | 65.1 | 57.1 |
| | | **KVCOMM** | **81.5** | **81.7** | **80.6** | **79.6** |
| | Reuse Rate (%) | Original | 0 | 0 | 0 | 0 |
| | | CacheBlend | 80 | 80 | 80 | 80 |
| | | **KVCOMM** | **79.6** | **77.0** | **73.4** | **71.0** |
| HumanEval | Pass@1 (%) | Original | 86.3 | 83.9 | 84.5 | 85.1 |
| | | CacheBlend | 31.1 | 21.1 | 30.4 | 32.9 |
| | | **KVCOMM** | **81.4** | **83.2** | **83.2** | **83.2** |
| | Reuse Rate (%) | Original | 0 | 0 | 0 | 0 |
| | | CacheBlend | 80 | 80 | 80 | 80 |
| | | **KVCOMM** | **87.6** | **84.7** | **81.1** | **77.8** |

sample's embedding proximity. Formally, the KV-cache of the $i$-th placeholder in the $m$-th agent is approximated as follows:

$$(\hat{\mathbf{k}}/\hat{\mathbf{v}})_{\phi_{(m,i)}} = (\mathbf{k}/\mathbf{v})_{\phi_{(m,i)}} + \sum_{\psi \in \mathcal{A}_{\phi_{(m,i)}}} w_{\phi_{(m,i)} \to \psi} \cdot \Delta(\mathbf{k}/\mathbf{v})^{\phi}_{(m,\psi)}, \tag{6}$$

where $(\hat{\mathbf{k}}/\hat{\mathbf{v}})_{\phi_{(m,i)}}$ refers to the approximated K/V cache for the placeholder $\phi_{(m,i)}$. $(\mathbf{k}/\mathbf{v})_{\phi_{(m,i)}}$ is the base K/V cache for the placeholder $\phi_{(m,i)}$. $w_{\phi_{(m,i)} \to \psi}$ is the softmax mapping of $-\|\mathbf{h}_{\phi_{(m,i)}} - \mathbf{h}_{\psi}\|$ across the anchor dimension. $\Delta(\mathbf{k}/\mathbf{v})^{\phi}_{(m,\psi)}$ is the placeholder $\phi$'s cache offsets in the $m$-th agent for the anchor $\psi$. Prefix segment updates follow an analogous process:

$$(\hat{\mathbf{k}}/\hat{\mathbf{v}})_{\mathbf{p}_{(m,i)}} = (\mathbf{k}/\mathbf{v})_{\mathbf{p}_{(m,i)}} + \sum_{\psi \in \mathcal{A}_{\phi_{(m,i)}}} w_{\phi_{(m,i)} \to \psi} \cdot \Delta(\mathbf{k}/\mathbf{v})^{\mathbf{P}}_{(m,\psi)}, \tag{7}$$

where $(\hat{\mathbf{k}}/\hat{\mathbf{v}})_{\mathbf{p}_{(m,i)}}$ refers to the approximated K/V cache for the prefix segment $\mathbf{p}_{(m,i)}$. $(\mathbf{k}/\mathbf{v})_{\mathbf{p}_{(m,i)}}$ is the base K/V cache for the prefix segment $\mathbf{p}_{(m,i)}$. $\Delta(\mathbf{k}/\mathbf{v})^{\mathbf{P}}_{(m,\psi)}$ is the corresponding prefix segment's cache offset in the $m$-th agent for the anchor $\psi$.

After approximation, updated caches are concatenated and directly fed into decoding, substantially reducing prefilling latency via parallel processing. The overall algorithm is listed in Appendix 6.2.5.

## 4 Experiments

### 4.1 Experimental Setup

**Multi-agent System.** Following GPTSwarm [69] and AgentPrune [65], we construct a fully-connected multi-agent system with established techniques including few-shot prompting [2], chain-of-thought [45], function calling [33], and structured outputs [34]. To precisely analyze KV-cache behaviors, we deploy open-source models using HuggingFace's framework rather than closed-source APIs used by AgentPrune. Specifically, we employ Llama-3.1-8B-Instruct [11] (Llama-3.1) for retrieval-augmented generation (RAG) and math reasoning, and Qwen-Coder-2.5-7B-Instruct [16] for programming tasks. We evaluate performance across scenarios ranging from two to five agents.

**Benchmark Datasets.** We assess RAG performance using MMLU [13], math reasoning with GSM8K [7], and programming capability via HumanEval [4].

Table 2: Per-agent TTFT breakdown and speedup. (Prefix token length per agent: 512; Output token length: 512; Model: Llama-3.1; #Agents = 5)

| TTFT (ms) | Agent 1 | Agent 2 | Agent 3 | Agent 4 | Agent 5 |
|---|---|---|---|---|---|
| Original | 125.8 | 192.4 | 258.3 | 330.9 | 428.6 |
| KVCOMM | 5.5 | 7.7 | 10.2 | 13.5 | 17.5 |
| 1st Token Decode | 21.4 | 21.2 | 21.2 | 21.3 | 21.1 |
| Others | 86.6 | 9.1 | 10.7 | 13.5 | 16.2 |
| **Speedup** | **1.11x** | **5.06x** | **6.14x** | **6.85x** | **7.82x** |

Table 3: Mean TTFT speedup using Llama-3.1. (#Agents = 3)

| Out_len | In_len (Prefix sequence) | | | | |
|---|---|---|---|---|---|
| | 64 | 128 | 256 | 512 | 1024 |
| **128** | 2.24x | 2.31x | 2.56x | 3.07x | 4.45x |
| **256** | 2.50x | 2.51x | 2.83x | 3.44x | 4.75x |
| **512** | 3.05x | 3.18x | 3.49x | 4.09x | 5.34x |
| **1024** | 4.40x | 4.48x | 4.75x | 5.35x | 6.72x |

**Comparison Baselines.** To our knowledge, KVCOMM is the first method enabling comprehensive KV-cache sharing tailored for open-source multi-agent frameworks. Hence, we compare primarily against CacheBlend [58], which selectively recomputes sensitive tokens' caches for partial reuse. Given CacheBlend's tight integration with vLLM [21], we faithfully replicate its selective recomputation strategy within our experimental setup, consistently recomputing the top-20% tokens exhibiting the largest KV deviations, ensuring fair baseline alignment.

**Evaluation Metrics.** We report Accuracy scores on MMLU and GSM8K, and Pass@1 for coding tasks. Efficiency metrics include Reuse Rate (meaning the proportion of agents employing the cache reuse scheme), individual agent Time-To-First-Token (TTFT), and average TTFT across agents.

**Implementation Details.** Experiments are executed on a single NVIDIA H100 GPU. The maximum generation length is uniformly set to 512 tokens, with hyperparameters selected as $\gamma = 0.3$ and anchor pool size $\mathcal{V} = 20$. Further implementation specifics are detailed in Appendix 6.3.

## 4.2 Main Results

Table 1 compares our KVCOMM approach with the Original (no cache reuse) and CacheBlend [58] strategies across multiple agent configurations. Although our agent prompts were initially optimized for closed-source models, causing some performance drops with open-source deployments, KVCOMM still maintains or improves upon baseline accuracy.

On MMLU, KVCOMM achieves competitive accuracy (64.7%–69.9%), consistently outperforming or matching CacheBlend and closely tracking the original baseline. This indicates robust cross-context KV-cache alignment by KVCOMM, CacheBlend fluctuates significantly with increasing agents. The reason why the baseline method performs poorly on the MMLU benchmark under the two-agent setting is that the first agent is designed as the *knowledgeable expert* to produce the related key words about the user question, and the second agent is designed as the *Final Refer* to analyze the predecessor agents' output and give the answer based on the previous agent's output, where the failure cases mainly occur when the second agent only output the answer without analysis. The prompt of each agent can be found in Appendix 6.3.2.

For GSM8K math reasoning, KVCOMM's accuracy remains stable, only declining by 1.9% (81.5%→79.6%) from two to five agents, maintaining within ±2% of the original baseline. In contrast, CacheBlend's accuracy drops dramatically from 82.0% to 57.1%, highlighting the necessity for precise KV-cache reuse in numerical tasks.

In the HumanEval coding benchmark, KVCOMM delivers stable Pass@1 scores (81.4%–83.2%), significantly surpassing CacheBlend by an average margin of 53%. This underscores KVCOMM's ability to preserve task-critical dependencies essential for programming tasks. The severe performance degradation of CacheBlend on Humaneval attributes to the diverse syntax separators involved in the generation process (e.g., ., ;, !), which induce diverse and prefix-sensitive KV-cache distributions.

**Reuse Rate.** Unlike CacheBlend's fixed reuse strategy (80%), KVCOMM adaptively determines KV-cache reuse, consistently achieving high reuse rates (70%–87.6%). This rate naturally declines as agent number increases due to more diverse contexts, but KVCOMM still effectively identifies shareable caches, confirming that adaptive reuse avoids context degradation.

## 4.3 Results of TTFT Speedup[1]

Table 2 reports TTFT per agent receiving 1K tokens from user input with 512 prefix tokens and sharing the 512 response tokens with succeeding agents. The first agent, lacking upstream caches

---

[1]In the original submission, the TTFT calculation for KVCOMM omitted the first token's decoding latency. The final version rectifies this.

(costing 86.6ms in "other" operations), shows modest acceleration ($1.11\times$). Subsequent agents reduce prefilling dramatically to 26.9–38.6 ms via KVCOMM, achieving up to **7.82$\times$** speedup (Agent 5).

**Scalability in Context Length.** We further examine scalability in Table 3, varying prefix (64–1K tokens) and output lengths (128–1K tokens) among three collaborating agents. KVCOMM achieves a minimum mean speedup of $2.24\times$ (shortest setting) and scales effectively to $6.72\times$ (longest setting), validating the approach's efficiency gain as context length and complexity increase.

## 4.4 Discussion and Ablation Study

**Robustness to Request Order.** Table 4 examines how request ordering affects KV-COMM's cache alignment using MMLU. We test two random orders (Rand-1, Rand-2), ascending and descending length orders. It can be observed that performance is correlated with request order due to the designed anchor prediction criterion. Results confirm KV-COMM is robust across diverse ordering strategies, achieving consistent or slightly improved accuracy compared to the baseline, demonstrating minimal sensitivity to request sequence variability.

Table 4: Study on the robustness to varying request orders. Accuracy is reported. (#Agent = 4, Baseline Acc = 68.0%; Model: Llama3.1)

| Method | Rand-1 | Rand-2 | Ascending | Descending |
|--------|--------|--------|-----------|------------|
| KVCOMM | 68.0 | 72.5 | 67.3 | 66.0 |

**Contribution of Each Alignment Step.** Table 5 details ablation results for three alignment components on MMLU under a four-agent setting: (1) position alignment via key rotation, (2) placeholder KV-cache offset, and (3) prefix segment KV-cache offset. The results reveal that each alignment step is critical; omitting any severely degrades accuracy. Although combining key rotation and prefix offset achieves 62.1% accuracy, the response of each agent is visibly less coherent with the original one (See Appendix 6.4.6). Therefore, complete alignment is essential for robust cross-context performance.

Table 5: Ablation study on MMLU under four-agent setting. (Model: Llama-3.1)

| $\mathbf{k}$ w/ rot | $\phi$ w/ offset | $\mathbf{p}$ w/ offset | Acc (%) |
|------|------|------|---------|
| ✓ | | | 43.1% |
| | ✓ | | 58.8% |
| | | ✓ | 60.1% |
| ✓ | ✓ | | 38.6% |
| ✓ | | ✓ | 62.1% |
| | ✓ | ✓ | 56.9% |
| ✓ | ✓ | ✓ | **68.0%** |

**Sensitivity to Hyperparameters.** Table 6 explores KVCOMM's sensitivity to the entropy threshold $\gamma$ and anchor pool size $\mathcal{V}$ using GSM8K with four agents. $\gamma = 0$ / $\mathcal{V} = 0$ refers to the original no-cache-sharing method. It can be observed that with conservative reuse ($\gamma = 0.1$), accuracy improves slightly (1%), while moderate relaxation significantly boosts reuse (up to 98.2%) at minimal accuracy cost (3.3%). For $\mathcal{V}$, performance is relatively stable with the increase of stored anchors, while the reuse rate finally becomes stable at 73.4%, indicating that $\mathcal{V} = 20$ effectively balances efficiency and task performance.

Table 6: Hyperparameter analysis on GSM8K under the four-agent setting using Llama-3.1.

| Metric | Threshold $\gamma$ ($\mathcal{V} = 20$) | | | | | |
|--------|------|------|------|------|------|------|
| | 0 | 0.1 | 0.3 | 0.5 | 0.7 | 0.9 |
| Accuracy (%) | 82.1 | 83.1 | **80.6** | 80.0 | 78.9 | 78.8 |
| Reuse rate (%) | N/A | 34.3 | **73.4** | 94.9 | 97.5 | 98.2 |

| Metric | Maximum Anchor Num $\mathcal{V}$ ($\gamma = 0.3$) | | | | | |
|--------|------|------|------|------|------|------|
| | 0 | 5 | 10 | 15 | 20 | 25 |
| Accuracy (%) | 82.1 | 82.0 | 81.4 | 81.2 | **80.6** | 80.6 |
| Reuse rate (%) | N/A | 44.0 | 60.3 | 66.2 | **73.4** | 73.4 |

## 5 Conclusion

In this paper, we explore KV-cache sharing for efficient communication in collaborative LLM-based MAS and introduce KVCOMM, a drop-in framework to enable efficient agent communication through shared KV-cache reuse and context-aware cache offsetting. Besides, we perform analyses of KV-cache deviation across varying prefix contexts, and propose an anchor-based offset estimator to effectively align and reuse shared context KV-caches. Extensive experiments conducted on Retrieval-Augmented Generation (RAG), Math Reasoning, and Programming-related multi-agent systems demonstrate that our method provides an effective trade-off between prefilling efficiency and system accuracy, continuously reducing average latency as the number of agents increases. Specifically, KVCOMM can achieve $\sim 6.7\times$ average prefilling speedup under the three-agent setting on a single H100 GPU, significantly improving the deployment efficiency of collaborative multi-agent language models.

# Acknowledgments

Hancheng Ye, Jianyi Zhang, and Yiran Chen disclose the support from NSF 2112562, ARO W911NF-23-2-0224, and NAIRR Pilot project NAIRR240270. Danyang Zhuo discloses the support from NSF 2503010. We sincerely thank the program chairs, area chair, and reviewers for their valuable comments.

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

# 6 Appendix

Due to the ten-page limitation of the manuscript, we provide more details and visualizations from the following aspects:

## 6.1 Limitations and Broader Impacts

Currently, KVCOMM is evaluated on LLM-based multi-agent systems that process text inputs. A long-term vision of multi-agent systems is to achieve lossless acceleration on any modality input, such as image, video, or audio input. Besides, although KVCOMM can accelerate the prefilling process of each agent, the decoding latency as another bottleneck for efficient collaboration between agents cannot be accelerated by KVCOMM, which will be the future work to co-optimize these two stages.

Furthermore, as mentioned in Sec. 3.1, KVCOMM is directly applicable for groups of homogeneous agents. For agents with identical architectures but different weights, KVCOMM holds promise but is pending further exploration. Also, in principle, KVCOMM can be applied wherever context change patterns recur, provided shared text segmentation is feasible (as with techniques similar to automatic prefix caching in vLLM). Although currently KVCOMM does not cover fully dynamic, unstructured cases, we recognize this as a meaningful direction and will extend KVCOMM for more dynamic agentic and debate-style benchmarks in future work. For the heterogeneous multi-agent systems with different attention formulations [25, 8], it remains underexplored how to use KV-cache communication to facilitate efficiency.

## 6.2 Method Explanation

### 6.2.1 Glossary

**Base KV-cache**  KV-cache for a prefix/placeholder under its initial context or without external input; serves as the reference for offsets.

**KV-cache offset / deviation**  The difference between a shared text's KV-cache under a new prefix and its base KV-cache *(Keys require RoPE-based alignment before offsetting)*.

**Placeholder / Prefix offset**  Offsets for the placeholder segment (external input) and its adjacent prefix segment (predefined context), respectively, relative to their base KV-caches.

**Offset variance problem**  KV-cache offsets for the same text can vary substantially across contexts, so static reuse is unreliable.

**Positional alignment / Key de-rotation**  RoPE de-rotation/re-rotation to align Keys before offsetting.

**Shareability**  Whether a request can skip prefill under anchor criteria; if not, dense prefilling is used.

**Shared memory / KV-cache**  The shared storage that holds *base* KV-caches across agents.

**Dense prefilling / generation**  The full computation path (prefill + decode) used when reuse is not possible.

**Anchor (pool)**  A small set of representative examples, each storing placeholder and prefix offsets, used to interpolate offsets for new contexts.

### 6.2.2   Theoretical Proof of Proposition 1 and 2

Throughout the proofs, we reuse the per-layer update defined in Eq. (2):

$$\mathbf{h}_n^{l+1} = \mathbf{h}_n^l + \text{FFN}^l\big(\mathbf{h}_n^l + \text{Attn}^l(\mathbf{h}^l)_n\big). \tag{A.1}$$

We introduce the Lipschitz conditions for attention and FFN modules [20, 43, 44, 41]:

$$\| \text{Attn}^l(\mathbf{h}^l)_n - \text{Attn}^l(\tilde{\mathbf{h}}^l)_n\| \le \alpha^l \sum_{i=1}^n \|\mathbf{h}_i^l - \tilde{\mathbf{h}}_i^l\|, \tag{A.2}$$

$$\| \text{FFN}^l(\mathbf{h}_j^l) - \text{FFN}^l(\tilde{\mathbf{h}}_j^l)\| \le \beta^l \|\mathbf{h}_j^l - \tilde{\mathbf{h}}_j^l\|. \tag{A.3}$$

**Proof of Proposition 1**  Define the maximum divergence between two hidden states $\mathbf{h}^l, \tilde{\mathbf{h}}^l$ at layer $l$:

$$\Delta^l = \max_{k \le n} \|\mathbf{h}_k^l - \tilde{\mathbf{h}}_k^l\|, \quad \text{with } \Delta^1 = \delta_n. \tag{A.4}$$

Subtracting two instances of Eq. (A.1) and applying Eq. (A.2) yields:

$$\|\mathbf{h}_n^{l+1} - \tilde{\mathbf{h}}_n^{l+1}\| \le \|\mathbf{h}_n^l - \tilde{\mathbf{h}}_n^l\| + \beta^l \left( \|\mathbf{h}_n^l - \tilde{\mathbf{h}}_n^l\| + \| \text{Attn}^l(\mathbf{h}^l)_n - \text{Attn}^l(\tilde{\mathbf{h}}^l)_n\| \right) \tag{A.5}$$

$$\le (1+\beta^l)\Delta^l + \beta^l \alpha^l \sum_{i=1}^n \|\mathbf{h}_i^l - \tilde{\mathbf{h}}_i^l\| \tag{A.6}$$

$$\le (1 + \beta^l + n\beta^l\alpha^l)\Delta^l. \tag{A.7}$$

Thus, we derive:

$$\Delta^{l+1} \le (1+\sigma^l)\Delta^l, \quad \text{where } \sigma^l = (1+n)\beta^l\alpha^l. \tag{A.8}$$

Unrolling this recursion from layer 1 to $l$, we have:

$$\Delta^l \le \delta_n \prod_{j=1}^{l-1}(1+\sigma^j). \tag{A.9}$$

Projecting to the key space via linear mapping $W_K^l$ and RoPE, whose spectral norms are bounded by constants $C_K^l$ and $C_R$ respectively, we have:

$$\|\mathbf{k}_n^l - \tilde{\mathbf{k}}_n^l\| \le C_R C_K^l \delta_n \prod_{j=1}^{l-1}(1+\sigma^j), \tag{A.10}$$

which completes the proof of Proposition 1. Due to analogous reasoning, the bound for value vectors is similar except for the absence of RoPE projection.

**Proof of Proposition 2**  Let two different prompts $\mathbf{p}_a$ and $\mathbf{p}_b$ be prefixed to both tokens $u$ and $v$. Denote their key vectors at positions $n_a$ when prefixed by $\mathbf{p}_a$ as $\mathbf{k}_{n_a}^l, \tilde{\mathbf{k}}_{n_a}^l$ for token $u$ and $v$ respectively, and at positions $n_b$ when prefixed by $\mathbf{p}_b$ as $\bar{\mathbf{k}}_{n_b}^l, \tilde{\bar{\mathbf{k}}}_{n_b}^l$ respectively. Define the two key vector deviations as follows:

$$\Delta^l = \bar{\mathbf{k}}_{n_b}^l - \mathbf{k}_{n_a}^l, \quad \tilde{\Delta}^l = \tilde{\bar{\mathbf{k}}}_{n_b}^l - \tilde{\mathbf{k}}_{n_a}^l. \tag{A.11}$$

Then we have:

$$\|\Delta^l - \tilde{\Delta}^l\| = \|\bar{\mathbf{k}}_{n_b}^l - \mathbf{k}_{n_a}^l - (\tilde{\bar{\mathbf{k}}}_{n_b}^l - \tilde{\mathbf{k}}_{n_a}^l)\| \tag{A.12}$$

$$\leq \|\mathbf{k}_{n_a}^l - \tilde{\mathbf{k}}_{n_a}^l\| + \|\bar{\mathbf{k}}_{n_b}^l - \tilde{\bar{\mathbf{k}}}_{n_b}^l\|. \tag{A.13}$$

Applying Proposition 1 to each term, we have:

$$\|\mathbf{k}_{n_a}^l - \tilde{\mathbf{k}}_{n_a}^l\| \leq C_R C_K^l \delta_{n_a} \prod_{j=1}^{l-1}(1 + \sigma^j), \tag{A.14}$$

$$\|\bar{\mathbf{k}}_{n_b}^l - \tilde{\bar{\mathbf{k}}}_{n_b}^l\| \leq C_R C_K^l \delta_{n_b} \prod_{j=1}^{l-1}(1 + \sigma^j), \tag{A.15}$$

where $\delta_{n_a} = \max_{k \leq n_a} \|\mathbf{h}_k^1 - \tilde{\mathbf{h}}_k^1\|$, $\delta_{n_b} = \max_{k \leq n_b} \|\bar{\mathbf{h}}_k^1 - \tilde{\bar{\mathbf{h}}}_k^1\|$. Since $\mathbf{h}_k^1 = \tilde{\mathbf{h}}_k^1$ for $k \leq n_a - 1$ (the same prefix $\mathbf{p}_a$), $\bar{\mathbf{h}}_k^1 = \tilde{\bar{\mathbf{h}}}_k^1$ for $k \leq n_b - 1$ (the same prefix $\mathbf{p}_b$), $\mathbf{h}_{n_a}^1 = \bar{\mathbf{h}}_{n_b}^1$ (the same token $u$), $\tilde{\mathbf{h}}_{n_a}^1 = \tilde{\bar{\mathbf{h}}}_{n_b}^1$ (the same token $v$), we have $\delta_{n_a} = \delta_{n_b} = \|\mathbf{h}_{n_a}^1 - \tilde{\mathbf{h}}_{n_a}^1\|$. Thus, we conclude:

$$\|\Delta^l - \tilde{\Delta}^l\| \leq 2 C_R C_K^l \delta_{n_a} \prod_{j=1}^{l-1}(1 + \sigma^j), \tag{A.16}$$

completing the proof of Proposition 2. The proof of the bound for value vectors is similar, except for the absence of RoPE projection.  □

Table A.1: KV-cache management strategy for both anchor pools and current requests' KV-cache sharing among agents.

| **Anchor Manager** | | | |
|---|---|---|---|
| | **1st level** | **2nd level** | **3rd level** |
| Indices | Placeholder ID, e.g., `user_question` | Anchor Index, e.g., `anchor[0]` | Agent ID / embedding, e.g., `agent_1_ph_Δ` |
| Values | Anchor List | `Dict` of different KV-cache offset in different agents and the anchor embedding tensor | KV-cache / embedding tensor |
| **Shared KV-cache Manager** | | | |
| | **1st level** | **2nd level** | **3rd level** |
| Indices | Agent ID / User Input, e.g., `agent_1` | Placeholder id, e.g., `response` | Turn Index, e.g., `response[-1]` |
| Values | `Dict` of different agents' response KV and outside input KV | `Dict` of response and prefix KV-caches | KV-cache list |

### 6.2.3  Placeholder Naming Rules

The placeholder in each agent's prompt template in a multi-agent system can be divided into three categories: user input, tool execution results, and responses from other agents [48, 69, 65]. Consequently, we design the name of each placeholder in the prompt template according to its category.

**Algorithm 1:** Anchor-based KV-cache Communication in Multi-Agent Systems (KVCOMM)

---

**Input:** Agent set $\mathcal{M}$ with prompts containing placeholders $\{\phi_{(m,i)}\}$; Anchor pool capacity $\mathcal{V}$; Entropy threshold $\gamma$.

**Output:** Efficiently updated KV-caches for all agents and responses from all agents.

**foreach** *agent* $m \in \mathcal{M}$ **do**

    Extract placeholder tokens $\{\phi_{(m,i)}\}$ from prompt;

    Initialize anchor pool $\mathcal{A}_\phi$ for each placeholder $\phi_{(m,i)}$ if not exist;

**foreach** *new input sample* **do**

    **foreach** *agent* $m \in \mathcal{M}$ **do**

        **if** *any placeholder sample is not in the shared memory* **then**

            Compute the base KV-caches for the placeholder samples absent in the shared memory and store them in the shared memory;

        **if** *all placeholders in the template are predicted as shareable according to Eq.* (5) **then**

            `// Reuse Placeholder KV-caches`

            **async foreach** *placeholder* $\phi_{(m,i)}$ *in agent* $m$ **do**

                `// Anchor Matching and Offset Approximation`

                Retrieve base KV-cache in the shared memory; Retrieve anchor pool $\mathcal{A}_{\phi_{(m,i)}}$;

                Identify matched anchors $\psi \in \mathcal{A}_{\phi_{(m,i)}}$;

                Compute weights $w_{\phi_{(m,i)} \to \psi} = \texttt{softmax}(-\|\mathbf{h}_{\phi_{(m,i)}} - \mathbf{h}_\psi\|)$ across anchors;

                Approximate KV-cache using Eq. (6);

                Similarly, update neighboring prefix segments using Eq. (7);

            Concatenate all updated $\{(\hat{\mathbf{k}}/\hat{\mathbf{v}})_{\phi_{(m,i)}}, (\hat{\mathbf{k}}/\hat{\mathbf{v}})_{\mathbf{P}_{(m,i)}}\}$ for agent $m$;

            Response and its KV-cache$\leftarrow$Decoding based on the concatenated KV-cache;

            **if** *the response's KV-cache is reusable according to Eq.* (5) **then**

                Store the response KV-cache in the shared memory for reference of other agents;

            **else**

                Store the response KV-cache in the anchor pool of the response placeholder as the base KV-cache of a new anchor;

        **else**

            `// Add as a new anchor`

            Response, Real KV-cache of all placeholders$\leftarrow$Dense generation for the input sample;

            **async foreach** *placeholder* $\phi_{(m,i)}$ *in agent* $m$ **do**

                Retrieve the base KV-cache of $\phi_{(m,i)}$ from the shared memory;

                $\Delta(\mathbf{k}/\mathbf{v})^\phi_{(m,\phi_{(m,i)})} \leftarrow$the offset between the real KV-cache and the base KV-cache of $\phi_{(m,i)}$;

                $\Delta(\mathbf{k}/\mathbf{v})^{\mathbf{P}}_{(m,\phi_{(m,i)})} \leftarrow$the offset between the real KV-cache and the base KV-cache of $\mathbf{P}_{(m,i)}$;

                $\mathcal{A}_{\phi_{(m,i)}} \leftarrow \mathcal{A}_{\phi_{(m,i)}} \cup \left\{ (\phi_{(m,i)}, (\mathbf{k}/\mathbf{v})_{\phi_{(m,i)}}, \Delta(\mathbf{k}/\mathbf{v})^\phi_{(m,\phi_{(m,i)})}, \Delta(\mathbf{k}/\mathbf{v})^{\mathbf{P}}_{(m,\phi_{(m,i)})}) \right\}$;

                **if** $|\mathcal{A}_{\phi_{(m,i)}}| > \mathcal{V}$ **then**

                    Prune least-frequently-used among earliest anchors in $\mathcal{A}_{\phi_{(m,i)}}$;

---

Suppose an agent is assigned a unique agent id as `xxx`, and is succeeded to the other two agents, whose agent id are `yyy` and `zzz` respectively, the naming rule of the placeholders in Agent `xxx`'s prompt template is defined as follows:

- User input: `{user_question}`;

- Tool execution results at the current turn: `{condition_xxx_current}`;

- Tool execution results at the previous $t$-th turn: `{condition_xxx_history_t}`;

- Response from Agent `yyy` at the current turn: `{agent_yyy_current}`;

- Response from Agent `zzz` at the current turn: `{agent_zzz_current}`;
- Response from itself at the previous $t$-th turn: `{agent_xxx_history_t}`;
- Response from Agent `yyy` at the previous $t$-th turn: `{agent_yyy_history_t}`;
- Response from Agent `zzz` at the previous $t$-th turn: `{agent_zzz_history_t}`;

Based on the above rule, it helps easily insert potential placeholders in each agent's initial prompt template and unify the addressing rule for all agents' placeholders in the shared anchor pools.

### 6.2.4 KV-cache Management Strategy

Regarding the cache management strategy in KVCOMM, we designed two three-level cache managers to achieve efficient writing and retrieving anchors' KV-caches and the current shared KV-caches, respectively. As shown in Table A.1, based on these two cache managers, each agent can quickly retrieve their intended KV-caches and also store their generated KV-caches. Meanwhile, we conduct the process of KV-cache storage and retrieval asynchronously, thus further improving the efficiency.

### 6.2.5 Algorithm of KVCOMM

The specific details of KVCOMM are shown in Algorithm 1.

### 6.3 More Experimental Details

#### 6.3.1 Statistical Analysis of KV-cache Proximity and Offset Proximity

In Figure 4, we aim to evaluate the correlation between the KV-cache proximity and the token embedding proximity, as well as the correlation between the KV-cache offset proximity and the token embedding proximity. This evaluation is crucial to validate the effectiveness of our approach in accurately approximating the KV-cache offsets of tokens by the embedding-closest anchors. For Figure 4a and 4b, we randomly sample 4000 distinct vocabulary tokens and then select 300 token pairs that are closest in the embedding space to form three equally-sized distance bins ("near", "mid", and "far"). For all token pairs, we prepend them with the same prefix context, test their K/V distance between each other, and compute the Spearman correlation coefficients [36] between the token distance and their K/V distances. For Figure 4c and 4d, we adopt the same setting, further prepend each token with two different prefixes, and test the cache deviation distance between different tokens.

#### 6.3.2 Prompt Design on Three Benchmarks

We show the detailed prompt template of each agent that is deployed in our experiments, which follows GPTSwarm [69] and AgentPrune [65].

**MMLU** We cycle through the following roles for multi-agent reasoning:

- Knowledgeable Expert
- Wiki Searcher
- Critic
- Mathematician
- FinalRefer

Below, we show the prompt template for each role:

> **Knowledgeable Expert**
> You are a knowledgeable expert in question answering. Please give at most six key entities that need to be searched in wikipedia to solve the problem. Key entities that need to be searched are included between two '@' when output, for example: @catfish effect@, @broken window effect@, @Shakespeare@. If there is no entity in the question that needs to be searched in Wikipedia, you don't have to provide it. The task is: `{user_question}`

**Wiki Searcher**

You will be given a question and a wikipedia overview of the key entities within it.

Please refer to them step by step to give your answer.

And point out potential issues in other agent's analysis.

The task is: {user_question}

The key entities of the problem are explained in Wikipedia as follows: {condition_1_current}

At the same time, the outputs of other agents are as follows:

Agent 1, role is Knowledge Expert, output is:

{agent_1_current}

In the last round of dialogue, the outputs of other agents were:

Agent 1, role is Knowledge Expert, output is:

{agent_1_history_-1}

---

**Critic**

You are an excellent critic.

Please point out potential issues in other agent's analysis point by point.

The task is: {user_question}

At the same time, the outputs of other agents are as follows:

Agent 1, role is Knowledge Expert, output is:

{agent_1_current}

Agent 2, role is Wiki Searcher, output is:

{agent_2_current}

In the last round of dialogue, the outputs of other agents were:

Agent 1, role is Knowledge Expert, output is:

{agent_1_history_-1}

Agent 2, role is Wiki Searcher, output is:

{agent_2_history_-1}

---

**Mathematician**

You are a mathematician who is good at math games, arithmetic calculation, and long-term planning.

The task is: {user_question}

At the same time, the outputs of other agents are as follows:

Agent 1, role is Knowledge Expert, output is:

{agent_1_current}

Agent 2, role is Wiki Searcher, output is:

{agent_2_current}

Agent 3, role is Critic, output is:

{agent_3_current}

In the last round of dialogue, the outputs of other agents were:

Agent 1, role is Knowledge Expert, output is:

{agent_1_history_-1}

Agent 2, role is Wiki Searcher, output is:

{agent_2_history_-1}

Agent 3, role is Critic, output is:

{agent_3_history_-1}

---

**FinalRefer**

You are the top decision-maker and are good at analyzing and summarizing other people's opinions, finding errors and giving final answers. You will receive a question followed by four possible answers labeled A, B, C, and D. Only one answer is correct. Choose the correct option based on the analysis and recommendations provided by the output of other agents. Your response must be exactly one of the letters A, B, C, or D, with no additional characters or text.

The task is: {user_question}

At the same time, the outputs of other agents are as follows:

Agent 1, role is Knowledge Expert, output is:

{agent_1_current}

Agent 2, role is Wiki Searcher, output is:

{agent_2_current}

Agent 3, role is Critic, output is:

```
{agent_3_current}
Agent 4, role is Mathematician, output is:
{agent_4_current}
In the last round of dialogue, the outputs of other agents were:
Agent 1, role is Knowledge Expert, output is:
{agent_1_history_-1}
Agent 2, role is Wiki Searcher, output is:
{agent_2_history_-1}
Agent 3, role is Critic, output is:
{agent_3_history_-1}
Agent 4, role is Mathematician, output is:
{agent_4_history_-1}
```

**GSM8K**  We cycle through the following roles:

- Math Solver
- Mathematical Analyst
- Programming Expert
- Inspector
- FinalRefer

Below are the prompt templates for each agent:

---

**Math Solver**
You are a math expert.
You will be given a math problem and hints from other agents.
Give your own solving process step by step based on hints.
The last line of your output contains only the final result without any units, for example: The answer is 140
You will be given some examples you may refer to. Q: Angelo and Melanie want to plan how many hours over the next week they should study together for their test next week.
They have 2 chapters of their textbook to study and 4 worksheets to memorize.
They figure out that they should dedicate 3 hours to each chapter of their textbook and 1.5 hours for each worksheet.
If they plan to study no more than 4 hours each day, how many days should they plan to study total over the next week if they take a 10-minute break every hour, include 3 10-minute snack breaks each day, and 30 minutes for lunch each day?.

A: Let's think step by step.
Angelo and Melanie think they should dedicate 3 hours to each of the 2 chapters, 3 hours x 2 chapters = 6 hours total.
For the worksheets they plan to dedicate 1.5 hours for each worksheet, 1.5 hours x 4 worksheets = 6 hours total. Angelo and Melanie need to start with planning 12 hours to study, at 4 hours a day, 12 / 4 = 3 days.
However, they need to include time for breaks and lunch. Every hour they want to include a 10-minute break, so 12 total hours x 10 minutes = 120 extra minutes for breaks.
They also want to include 3 10-minute snack breaks, 3 x 10 minutes = 30 minutes.
And they want to include 30 minutes for lunch each day, so 120 minutes for breaks + 30 minutes for snack breaks + 30 minutes for lunch = 180 minutes, or 180 / 60 minutes per hour = 3 extra hours.
So Angelo and Melanie want to plan 12 hours to study + 3 hours of breaks = 15 hours total.
They want to study no more than 4 hours each day, 15 hours / 4 hours each day = 3.75
They will need to plan to study 4 days to allow for all the time they need.
The answer is 4

Q: Bella has two times as many marbles as frisbees. She also has 20 more frisbees than deck cards. If she buys 2/5 times more of each item, what would be the total number of the items she will have if she currently has 60 marbles?
A: Let's think step by step
When Bella buys 2/5 times more marbles, she'll have increased the number of marbles by 2/5*60 = 24
The total number of marbles she'll have is 60+24 = 84

---

If Bella currently has 60 marbles, and she has two times as many marbles as frisbees, she has 60/2 = 30 frisbees.
If Bella buys 2/5 times more frisbees, she'll have 2/5*30 = 12 more frisbees.
The total number of frisbees she'll have will increase to 30+12 = 42
Bella also has 20 more frisbees than deck cards, meaning she has 30-20 = 10 deck cards
If she buys 2/5 times more deck cards, she'll have 2/5*10 = 4 more deck cards.
The total number of deck cards she'll have is 10+4 = 14
Together, Bella will have a total of 14+42+84 = 140 items The answer is 140

Q: Susy goes to a large school with 800 students, while Sarah goes to a smaller school with only 300 students. At the start of the school year, Susy had 100 social media followers. She gained 40 new followers in the first week of the school year, half that in the second week, and half of that in the third week. Sarah only had 50 social media followers at the start of the year, but she gained 90 new followers the first week, a third of that in the second week, and a third of that in the third week. After three weeks, how many social media followers did the girl with the most total followers have?
A: Let's think step by step
After one week, Susy has 100+40 = 140 followers.
In the second week, Susy gains 40/2 = 20 new followers.
In the third week, Susy gains 20/2 = 10 new followers.
In total, Susy finishes the three weeks with 140+20+10 = 170 total followers.
After one week, Sarah has 50+90 = 140 followers.
After the second week, Sarah gains 90/3 = 30 followers.
After the third week, Sarah gains 30/3 = 10 followers.
So, Sarah finishes the three weeks with 140+30+10 = 180 total followers.
Thus, Sarah is the girl with the most total followers with a total of 180.
The answer is 180
Q: {user_question}

---

**Mathematical Analyst**
You are a mathematical analyst. You will be given a math problem, analysis and code from other agents. You need to first analyze the problem-solving process step by step, where the variables are represented by letters. Then you substitute the values into the analysis process to perform calculations and get the results. The last line of your output contains only the final result without any units, for example: The answer is 140 You will be given some examples you may refer to.
Q: There are 15 trees in the grove. Grove workers will plant trees in the grove today. After they are done, there will be 21 trees. How many trees did the grove workers plant today?
A: ## Problem solving process analysis
There are {ori_tree_num} trees originally.
Then there were {after_planted_tree_num} trees after some more were planted.
So the number of trees planted today {today_planted_num} is the number of trees after planting {after_planted_tree_num} minus the number of trees before planting {ori_tree_num}.
The answer is {today_planted_num} = {after_planted_tree_num} - {ori_tree_num}.
## Actual analysis and solution process
In this question, {ori_tree_num} = 15 and {after_planted_tree_num} = 21.
There are 15 trees originally.
Then there were 21 trees after some more were planted.
So the number of trees planted today must have been 21 - 15 = 6.
The answer is 6

Q: Leah had 32 chocolates and her sister had 42. If they ate 35, how many pieces do they have left in total?
A:## Problem solving process analysis
Originally, Leah had {Leah_num} Leah_num chocolates.
Her sister had {sister_num} chocolates.
So in total they had {all_num} = {Leah_num} + {sister_num} chocolates.
After eating {eating_num} chocolates, the number of chocolates they have left {remain_num} is {all_num} minus {eating_num}.
The answer is {remain_num} = {all_num} - {eating_num}.
## Actual analysis and solution process
In this question, {Leah_num} = 32, {sister_num} = 42 and {all_num} = 35.
So, in total they had 32 + 42 = 74 chocolates originally.

After eating 35 chocolates, they had 74 - 35 = 39 chocolates.
The answer is 39
Q: {user_question}
At the same time, there are the following responses to the same question for your reference:
Agent 1 as a Math Solver his answer to this question is:
{agent_1_current}
In the last round of dialogue, there were the following responses to the same question for your reference:
Agent 1 as a Math Solver his answer to this question was:
{agent_1_history_-1}

---

**Programming Expert**

You are a programming expert. You will be given a math problem, analysis and code from other agents. Integrate step-by-step reasoning and Python code to solve math problems. Analyze the question and write functions to solve the problem. The function should not take any arguments and use the final result as the return value. The last line of code calls the function you wrote and assigns the return value to the `answer` variable. Use a Python code block to write your response. For example:

```python
def fun():
    x = 10
    y = 20
    return x + y
answer = fun()
```

Do not include anything other than Python code blocks in your response. You will be given some examples you may refer to. Q: Olivia has $23. She bought five bagels for $3 each. How much money does she have left?
A:

```python
def money_left():
    money_initial = 23
    bagels = 5
    bagel_cost = 3
    money_spent = bagels * bagel_cost
    remaining_money = money_initial - money_spent
    return remaining_money
answer = money_left()
```

Q: Michael had 58 golf balls. On tuesday, he lost 23 golf balls. On wednesday, he lost 2 more. How many golf balls did he have at the end of wednesday?
A:

```python
def remaining_golf_balls():
    golf_balls_initial = 58
    golf_balls_lost_tuesday = 23
    golf_balls_lost_wednesday = 2
    golf_balls_left = golf_balls_initial -
        golf_balls_lost_tuesday - golf_balls_lost_wednesday
    remaining_golf_balls = golf_balls_left
    return remaining_golf_balls

answer = remaining_golf_balls()
```

Q: {user_question}
At the same time, there are the following responses to the same question for your reference:

Agent 1 as a Math Solver his answer to this question is:
`{agent_1_current}`
Agent 2 as a Mathematical Analyst his answer to this question is:
`{agent_2_current}`
In the last round of dialogue, there were the following responses to the same question for your reference:
Agent 1 as a Math Solver his answer to this question was:
`{agent_1_history_-1}`
Agent 2 as a Mathematical Analyst his answer to this question was:
`{agent_2_history_-1}`

---

**Inspector**
You are an Inspector. You will be given a math problem, analysis and code from other agents. Check whether the logic/calculation of the problem solving and analysis process is correct (if present). Check whether the code corresponds to the solution analysis (if present). Give your own solving process step by step based on hints. The last line of your output contains only the final result without any units, for example: The answer is 140 You will be given some examples you may refer to.
Q: `{user_question}`
At the same time, there are the following responses to the same question for your reference:
Agent 1 as a Math Solver his answer to this question is:
`{agent_1_current}`
Agent 2 as a Mathematical Analyst his answer to this question is:
`{agent_2_current}`
Agent 3 as a Programming Expert his answer to this question is:
`{agent_3_current}`
the result is `{condition_3_current}`
In the last round of dialogue, there were the following responses to the same question for your reference:
Agent 1 as a Math Solver his answer to this question was:
`{agent_1_history_-1}`
Agent 2 as a Mathematical Analyst his answer to this question was:
`{agent_2_history_-1}`
Agent 2 as a Programming Expert his answer to this question was:
`{agent_3_history_-1}`
the result is `{condition_3_history_-1}`

---

**FinalRefer**
You are the top decision-maker. Good at analyzing and summarizing mathematical problems, judging and summarizing other people's solutions, and giving final answers to math problems. You will be given a math problem, analysis and code from other agents. Please find the most reliable answer based on the analysis and results of other agents. Give reasons for making decisions. The last line of your output contains only the final result without any units, for example: The answer is 140
The task is: `{user_question}`
At the same time, the output of other agents is as follows:
Agent 1, role is Math Solver, output is:
`{agent_1_current}`
Agent 2, role is Mathematical Analyst, output is:
`{agent_2_current}`
Agent 3, role is Programming Expert, output is:
`{agent_3_current}`
the result is `{condition_3_current}`
Agent 4, role is Inspector, output is:
`{agent_4_current}`
In the last round of dialogue, the outputs of other agents were:
Agent 1, role is Math Solver, output is:
`{agent_1_history_-1}`

> Agent 2, role is Mathematical Analyst, output is:
> `{agent_2_history_-1}`
> Agent 3, role is Programming Expert, output is:
> `{agent_3_history_-1}`
> the result is `{condition_3_history_-1}`
> Agent 4, role is Inspector, output is:
> `{agent_4_history_-1}`

**HumanEval Benchmark**    For HumanEval, the following roles are cycled for collaborative code generation and debugging:

- Project Manager
- Algorithm Designer
- Programming Expert
- Test Analyst
- FinalRefer

Below are the prompt templates for each agent:

> **Project Manager**
> You are a project manager. You will be given a function signature and its docstring by the user. You are responsible for overseeing the overall structure of the code, ensuring that the code is structured to complete the task. Implement code concisely and correctly without pursuing over-engineering. You need to suggest optimal design patterns to ensure that the code follows best practices for maintainability and flexibility. You can specify the overall design of the code, including the classes that need to be defined (maybe none) and the functions used (maybe only one function). I hope your reply will be more concise. Preferably within fifty words. Don't list too many points.
> The task is: `{user_question}`

> **Algorithm Designer**
> You are an algorithm designer. You will be given a function signature and its docstring by the user. You need to specify the specific design of the algorithm, including the classes that may be defined and the functions used. You need to generate the detailed documentation, including explanations of the algorithm, usage instructions, and API references. When the implementation logic is complex, you can give the pseudocode logic of the main algorithm. I hope your reply will be more concise. Preferably within fifty words. Don't list too many points.
> The task is: `{user_question}`
> At the same time, the outputs and feedbacks of other agents are as follows:
> Agent 1 as a Project Manager:
> The code written by the agent is:
> `{agent_1_current}`
> Whether it passes internal testing?
> `{condition_1_current}`
> In the last round of dialogue, the outputs and feedbacks of some agents were:
> Agent 1 as a Project Manager:
> The code written by the agent is:
> `{agent_1_history_-1}`
> Whether it passes internal testing?
> `{condition_1_history_-1}`

> **Programming Expert**
> You are a programming expert. You will be given a function signature and its docstring by

the user. You may be able to get the output results of other agents. They may have passed
internal tests, but they may not be completely correct. Write your full implementation (restate
the function signature). Use a Python code block to write your response. For example:

```python
print('Hello world!')
```

Do not include anything other than Python code blocks in your response. Do not change
function names and input variable types in tasks.
The task is: {user_question}
At the same time, the outputs and feedbacks of other agents are as follows:
Agent 1 as a Project Manager:
The code written by the agent is:
{agent_1_current}
Whether it passes internal testing?
{condition_1_current}
Agent 2 as a Algorithm Designer provides the following info:
{agent_2_current}
In the last round of dialogue, the outputs and feedbacks of some agents were:
Agent 1 as a Project Manager:
The code written by the agent was:
{agent_1_history_-1}
Whether it passed internal testing?
{condition_1_history_-1}
Agent 2 as a Algorithm Designer provided the following info:
{agent_2_history_-1}

---

**Test Analyst**
You are a test analyst. You will be given a function signature and its docstring by the user. You
need to provide problems in the current code or solution based on the test data and possible
test feedback in the question. You need to provide additional special use cases, boundary
conditions, etc. that should be paid attention to when writing code. You can point out any
potential errors in the code. I hope your reply will be more concise. Preferably within fifty
words. Don't list too many points.
The task is: {user_question}
At the same time, the outputs and feedbacks of other agents are as follows:
Agent 1 as a Project Manager:
The code written by the agent is:
{agent_1_current}
Whether it passes internal testing?
{condition_1_current}
Agent 2 as a Algorithm Designer provides the following info:
{agent_2_current}
Agent 3 as a Programming Expert:
The code written by the agent is:
{agent_3_current}
Whether it passes internal testing?
{condition_3_current}
In the last round of dialogue, the outputs and feedbacks of some agents were:
Agent 1 as a Project Manager:
The code written by the agent was:
{agent_1_history_-1}
Whether it passed internal testing?
{condition_1_history_-1}
Agent 2 as an Algorithm Designer provided the following info:
{agent_2_history_-1}

Agent 3 as a Programming Expert:
The code written by the agent was:
{agent_3_history_-1}
Whether it passed internal testing?
{condition_3_history_-1}

**FinalRefer**
You are the top decision-maker and are good at analyzing and summarizing other people's opinions, finding errors, and giving final answers. And you are an AI that only responds with only Python code. You will be given a function signature and its docstring by the user. You may be given the overall code design, algorithm framework, code implementation or test problems. Write your full implementation (restate the function signature). If the prompt given to you contains code that passed internal testing, you can choose the most reliable reply. If there is no code that has passed internal testing in the prompt, you can change it yourself according to the prompt. Use a Python code block to write your response. For example:

```python
print('Hello world!')
```

Do not include anything other than Python code blocks in your response
The task is: {user_question}
At the same time, the output of other agents is as follows:
Agent 1 as a Project Manager:
The code written by the agent was:
{agent_1_current}
Whether it passed internal testing?
{condition_1_current}
Agent 2 as an Algorithm Designer provided the following info:
{agent_2_current}
Agent 3 as a Programming Expert:
The code written by the agent was:
{agent_3_current}
Whether it passed internal testing?
{condition_3_current}
Agent 4, role is Test Analyst, output is:
The code written by the agent was:
{agent_4_current}
Whether it passed internal testing?
{condition_4_current}
In the last round of dialogue, the outputs of other agents were:
Agent 1 as a Project Manager:
The code written by the agent was:
{agent_1_history_-1}
Whether it passed internal testing?
{condition_1_history_-1}
Agent 2 as an Algorithm Designer provided the following info:
{agent_2_history_-1}
Agent 3 as a Programming Expert:
The code written by the agent was:
{agent_3_history_-1}
Whether it passed internal testing?
{condition_3_history_-1}
Agent 4, role is Test Analyst, output is:
The code written by the agent was:
{agent_4_history_-1}
Whether it passed internal testing?
{condition_4_history_-1}

Table A.2: Performance on MATH500 and AIME under different numbers of collaborating agents. Higher is better for Accuracy and Reuse Rate.

| Dataset | Model | Method | # Agents | | |
|---------|-------|--------|:---:|:---:|:---:|
| | | | **2** | **3** | **4** |
| MATH500 | Llama-3.1 | Acc. (%) Original | 41.6 | 39.6 | 42.6 |
| | | **Acc. (%) KVComm** | **38.0** | **38.6** | **44.2** |
| | | Reuse Rate (%) Original | 0 | 0 | 0 |
| | | **Reuse Rate (%) KVComm** | **59.4** | **40.9** | **30.7** |
| | Deepseek-Qwen | Acc. (%) Original | 51.4 | 49.6 | 42.8 |
| | | **Acc. (%) KVComm** | **50.8** | **50.8** | **45.8** |
| | | Reuse Rate (%) Original | 0 | 0 | 0 |
| | | **Reuse Rate (%) KVComm** | **76.7** | **60.4** | **45.3** |
| AIME | Deepseek-Qwen | Acc. (%) Original | 19.2 | 17.5 | 17.5 |
| | | **Acc. (%) KVComm** | **11.7** | **10.8** | **8.3** |
| | | Reuse Rate (%) Original | 0 | 0 | 0 |
| | | **Reuse Rate (%) KVComm** | **71.3** | **78.1** | **74.6** |

## 6.4 More Experimental Analysis

In this section, we provide more analysis on KVComm, including the effectiveness of the proposed matching criterion, the effectiveness of the $\ell_2$ norm-based approximation method, and more visualizations for understanding the effectiveness and efficiency of KVComm.

### 6.4.1 Evaluations on Harder Reasoning Benchmarks

We further provide evaluation results on the MATH500 [14] and AIME [6] benchmarks to examine the effectiveness of KVComm under harder reasoning tasks. For MATH500, we tested both Llama3.1-8B-instruct [11] (Llama-3.1) and Deepseek-R1-Distill-Qwen-7B [12] (Deepseek-Qwen). As shown in Table A.2, KVComm achieves superior or comparable performance to dense computation.

Compared with relatively easy math reasoning tasks such as GSM8K, the reuse rate indeed drops more rapidly as the number of agents increases. Nevertheless, the accuracy of KVComm remains competitive or even surpasses the baseline, indicating that referencing anchors' information can consistently help. A notable insight from Deepseek-Qwen results is that KVComm achieves both a higher reuse rate and a higher accuracy on reasoning-oriented models.

We also evaluate on the more challenging AIME benchmark with Deepseek-Qwen. As shown in Table A.2, KVComm still maintains comparable accuracy to dense prefill while keeping the reuse rate above 70%. The accuracy drop here is mainly due to the token length constraint during decoding: since KVComm accelerates prefill at the cost of extra memory, the effective decoding length per agent is reduced.

### 6.4.2 Analysis on the Matching Criterion

In this section, we further examine the effectiveness of our proposed anchor matching criterion, evaluating the $\ell_2$ norm-based matching criterion. Table A.3 compares two distinct matching criteria for anchor prediction and the approximation of placeholder samples on the MMLU benchmark in the four-agent scenario. Recognizing that the length match is a fundamental requirement for effective token-level approximation within placeholder samples, we explicitly evaluate the contribution of the complementary distance-based matching criterion $\mathcal{H}_{\phi|\mathcal{A}}$. It is observed that omitting the distance-based criterion can increase anchor reuse rates; however, this is accompanied by a marked performance degradation, indicating

Table A.3: Ablation study of the matching criterion on MMLU under a four-agent setting. (Model: Llama-3.1)

| Method | Acc (%) | Reuse Rate (%) |
|--------|:---:|:---:|
| $\mathcal{L}_{\phi}$ | 62.1 | 93.3 |
| $\mathcal{L}_{\phi} \& \mathcal{H}_{\phi|\mathcal{A}}$ | **68.0** | **70.1** |

Table A.5: Simulated `softmax` latency (ms) with different anchor counts and sequence lengths.

| #Anchor | 1024-tokens | 2048-tokens | 4096-tokens |
|---------|-------------|-------------|-------------|
| 5 | 0.894 | 1.719 | 3.552 |
| 10 | 1.773 | 3.576 | 7.128 |
| 15 | 2.620 | 5.332 | 10.766 |
| 20 | 3.933 | 7.859 | 15.624 |
| 25 | 4.435 | 9.614 | 18.113 |

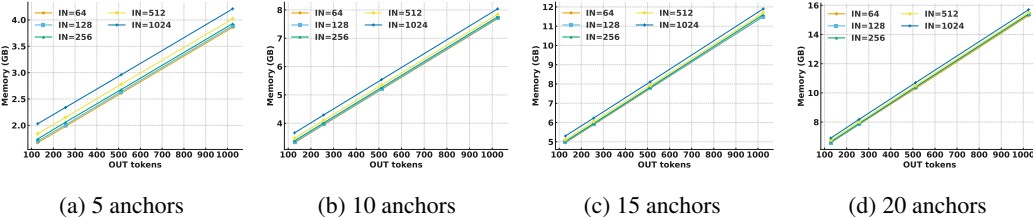

(a) 5 anchors     (b) 10 anchors     (c) 15 anchors     (d) 20 anchors

Figure A.1: Memory cost (GB) across IN/OUT for different anchor counts.

that mere length matching is insufficient for optimal anchor utilization. Conversely, integrating the embedding-distance criterion achieves an advantageous balance, maintaining high performance while providing effective anchor reuse. These results highlight the necessity of a comprehensive matching criterion that incorporates both structural alignment (length matching) and semantic similarity (embedding-distance criterion) to ensure both efficiency and task accuracy in KVCOMM.

### 6.4.3 Analysis on the Approximation Method

We further investigate the effectiveness of different KV-cache offset approximation methods on HumanEval under a four-agent setting (Table A.4). Two comparison methods are introduced: one using the nearest anchor's KV-cache offset based on the $\ell_2$ norm, and another employing cosine similarity with `softmax`-based anchor weighting. Results indicate the cosine-similarity-based method achieves performance com-

Table A.4: Performance of different approximation methods for KV-cache offsets on HumanEval under a four-agent setting. (Model: Qwen-Coder-2.5-7B, Baseline Acc: 84.45%)

| Method | Acc (%) | Reuse Rate (%) |
|--------|---------|----------------|
| Nearest | 47.20 | 78.9 |
| Cosine Similarity | 83.23 | 82.5 |
| **Ours ($\ell_2$ Norm)** | **83.23** | **81.1** |

parable to our $\ell_2$ norm-based method with slightly higher reuse rates. However, the nearest-reusing approach significantly deteriorates performance due to the error induced by the distance between the sample and the nearest anchor. Thus, soft aggregation of anchors proves to be an effective approximation method.

### 6.4.4 Analysis of the Overhead in Long-context Anchor Matching

In KVCOMM, `softmax` is operated along the anchor number dimension on the negative $\ell_2$-norm distance between the placeholder sample and each anchor, which reduces the Key/Value tensor into the shape of [$m$, 1, 1, seq_len, 1] ($m$: anchor count, seq_len: sequence length). The latency of softmax thus scales with both parameters. We quantify its latency overhead in Table A.5. Here the shape of each KV-cache tensor is [32, 8, seq_len, 128]. We can observe that latency remains reasonable ($\sim$18 ms with 25 anchors and 4096 tokens per anchor).

These results are from a simulation without competing system workloads. In real multi-agent long-context scenarios, the latency also includes offloading KV-caches to the CPU. For example, as shown in Table A.6 (using Llama3.1-8B-instruct on the MMLU benchmark), the average softmax latency is 100+ ms when all anchors are offloaded to CPU, while total offloading per agent can reach 1260+ ms for 4K-token contexts. This indicates that **the main overhead arises from data movement in the `softmax` operation for the long-context KV**. Such communication overhead can be mitigated with systematic optimization (e.g., pipelining), which is orthogonal to the KVCOMM mechanism.

Table A.6: Latency and memory cost of 4K-tokens anchor matching using Llama3.1-8B-instruct on the MMLU benchmark.

| Metrics | #Agent=3 | #Agent=4 |
|---|---|---|
| Softmax (ms) | 104 | 122 |
| Offloading (ms) | 1260 | 1300 |
| Acc. (%) | 66.7 | 68.0 |
| Reuse Rate (%) | 49.7 | 51.0 |
| Memory Cost (GB) | 68.5 | 95.1 |

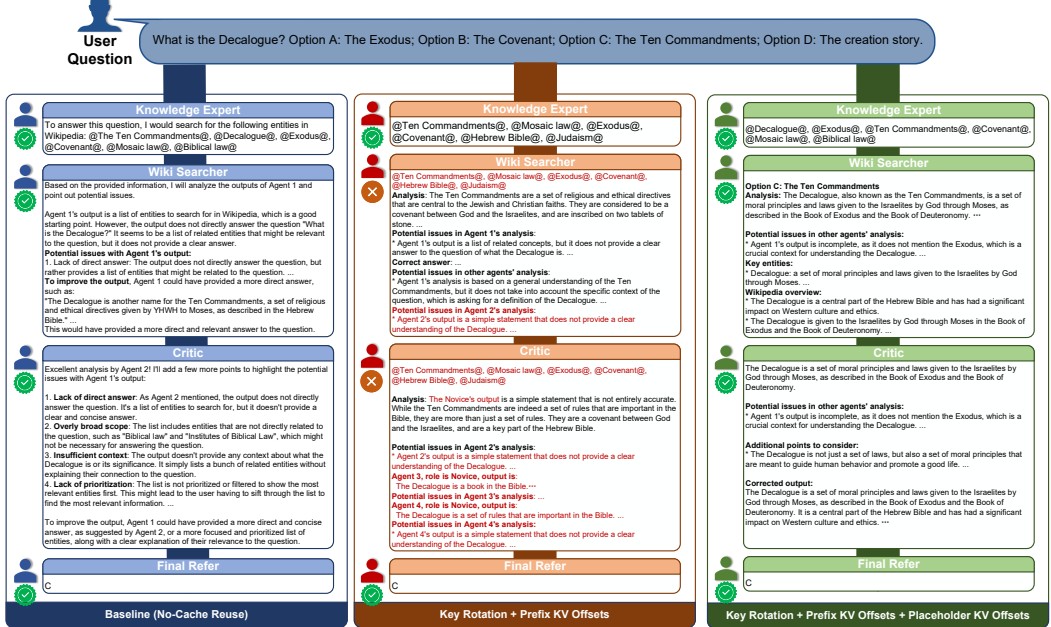

Figure A.2: Generation Comparison with different alignment combinations. **Left**: Original response of each agent without cache reuse. **Middle**: Responses generated by cache reuse, associated with aligning the position encoding of key caches and offsetting the prefix segments' caches. **Right**: Responses generated by combining all three alignment processes.

### 6.4.5 Analysis on the Memory Cost of KVCOMM

We also report memory overhead under different agentic configurations, varying input length (prefix), output length (response), and the number of anchors per placeholder. As shown in Figure A.1 (three-agent setting), the memory cost increases with longer IN/OUT sequences and with larger anchor counts, reflecting the storage required for anchor-specific KV-cache deviations. Empirically, we observe that these deviation tensors are quite sparse across anchors, with on average about $50\%$ of elements having absolute values smaller than $10^{-1}$. This indicates substantial headroom for *lossless* compression of anchors, which will be one of the future work to support longer contexts without sacrificing prefill speedups.

### 6.4.6 Visualization of Responses Generated by Different Combinations of Alignment Strategies

We further visualize the detailed response of each agent using different cache alignment strategies. As shown in Figure A.2, it can be observed that although the middle setting (Key Rotation + Prefix KV Offsets) eventually outputs the correct option "C", its reasoning chain is clearly degraded: the agents omit a formal definition of "Decalogue", repeat the keywords from the Knowledge Expert agent, and make analysis on agents that do not exist, resulting in a logically fragmented discourse. This "right answer for the wrong reasons" phenomenon underscores that two-level alignment alone

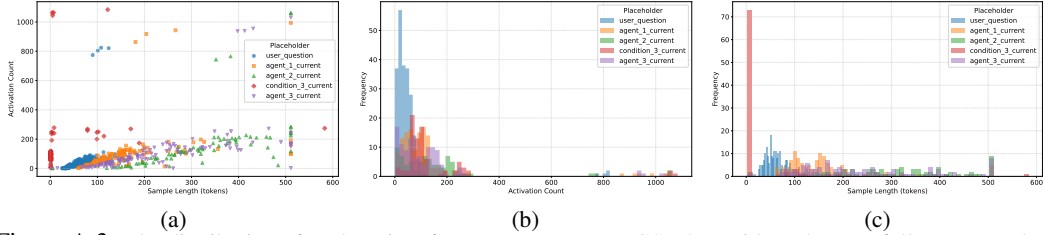

| (a) | (b) | (c) |

Figure A.3: The distribution of anchors in a four-agent system on GSM8K, with each agent fully connected to its predecessors within a single conversational turn. (a) Relationship between anchor activation counts and token lengths across placeholders during inference. (b) Histogram of anchor activation counts across all placeholders. (c) Histogram of anchor token lengths across all placeholders.

Table A.7: Differences between *placeholder offset* and *prefix offset*.

| Aspect | Placeholder offset | Prefix offset |
|---|---|---|
| **Definition** | KV-cache offset for *externally injected text* (users, agents, tools, etc.); its base KV-cache is not preset. | KV-cache offset for *predefined prompt text* (system prompt, placeholder conjunctions, suffix). |
| **Dependence** | Arises mainly from changes to the *prefix context*. | Triggered by changes to the *preceding placeholder* (injected content). |
| **Variance** | Shows *high variance* across samples and contexts. | Typically *more stable*, since base KV-caches are computed under a fixed system prompt and are less influenced by preceding placeholders. |

still disrupts the causal dependency between evidence, inference, and conclusion; the combination of all three alignment processes is therefore essential for achieving a coherent explanation path that is comparable to the original response.

### 6.4.7  Visualization of Anchor Distribution

As illustrated in Figure A.3, we visualize the anchor distribution for a four-agent system evaluated on the GSM8K dataset. Figure A.3a demonstrates a clear positive correlation between an anchor's token length and its activation frequency across conversational placeholders (*user_question* and three *agent_response placeholders*). Notably, the *condition_3_current* anchors, which are the execution results of the generated Python codes, exhibit a distinctive bimodal distribution: one group is extremely short but heavily reused (less than 10 tokens, activated over 1000 times), while another spans longer lengths with relatively sparse activations. Figure A.3b and A.3c further emphasize this long-tailed phenomenon, showing that the majority of anchors have activation counts under 100 and token lengths shorter than 200. This skewed distribution justifies our cache management strategy, prioritizing memory allocation for high-frequency anchors and dynamically pruning infrequently reused ones.

### 6.4.8  Visualization of Difference between Prefix and Placeholder Offset Distributions

To clarify the difference between prefix and placeholder offsets, we describe them from three aspects, which are shown in Table A.7.

We further visualize the offset distributions of these two types, which is experimented in a fully-connected four-agent setting on the MMLU dataset. We tested the offset variance among ten different samples, and present them in Figure A.4, A.5, A.6, A.7, A.8, A.9, A.10, and A.11. It can be observed that while the offset of the prefix KV is often larger than the placeholder one, its variance is relatively smaller than the placeholder KV, especially in deep layers (e.g., Figure A.5). The reason is that: During the precomputation of prefix KV-caches, the subsequent prefix segments are primarily correlated with the first prefix segment, typically containing system prompts; thus, their base KV-caches are relatively stable.

### 6.4.9  Visualization of Distance between Approximated and Real Offsets

Figure A.12 further compares various approximation strategies on HumanEval using Qwen-Coder-2.5-7B for the four-agent scenario, focusing on the similarity and error between approximated and real KV-caches. Overall, our proposed $\ell_2$-norm-based (L2NORM) approximation exhibits

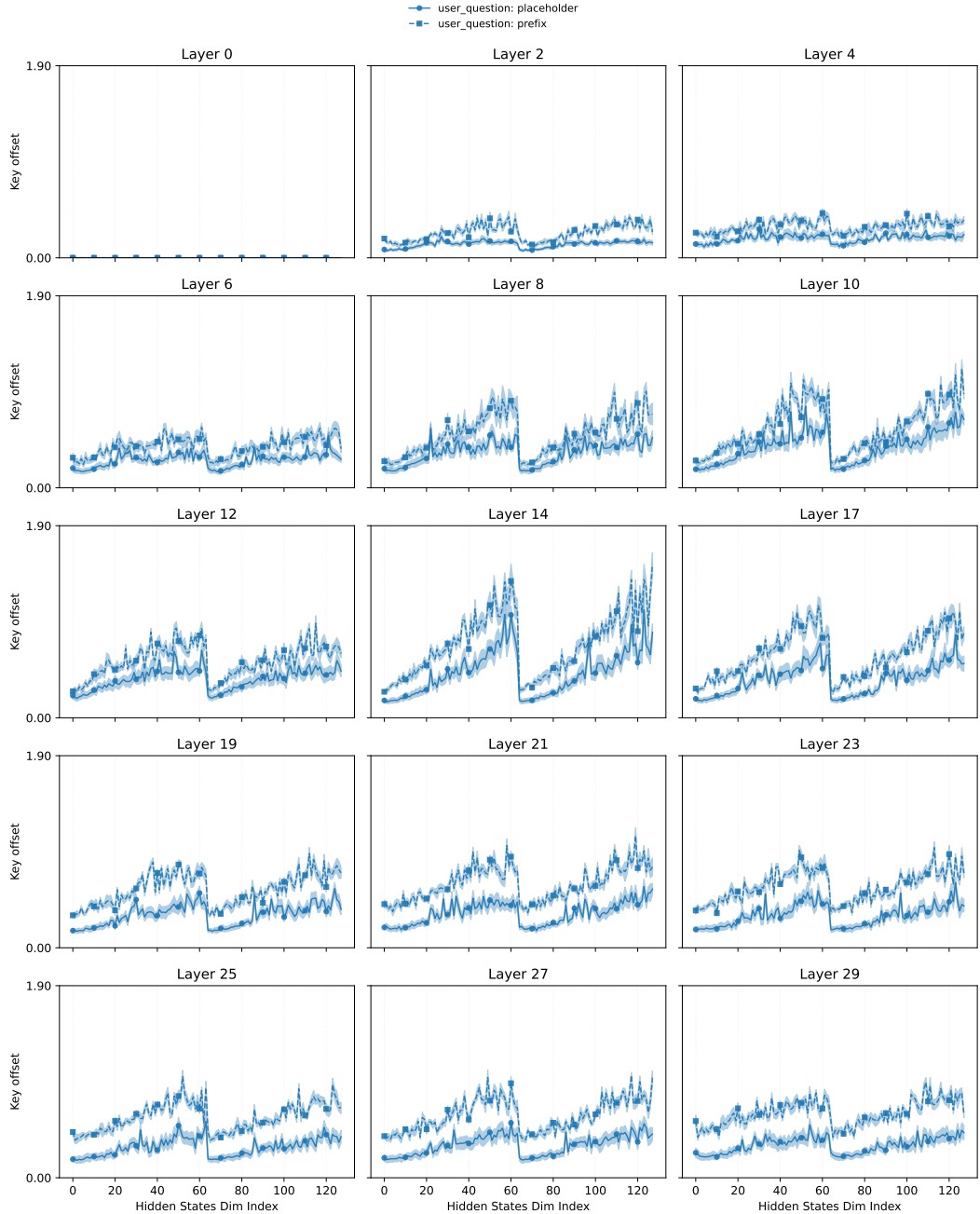

Figure A.4: Key cache offset distributions of the first agent's placeholder and prefix segments in a four-agent setting on the ten samples from the MMLU dataset.

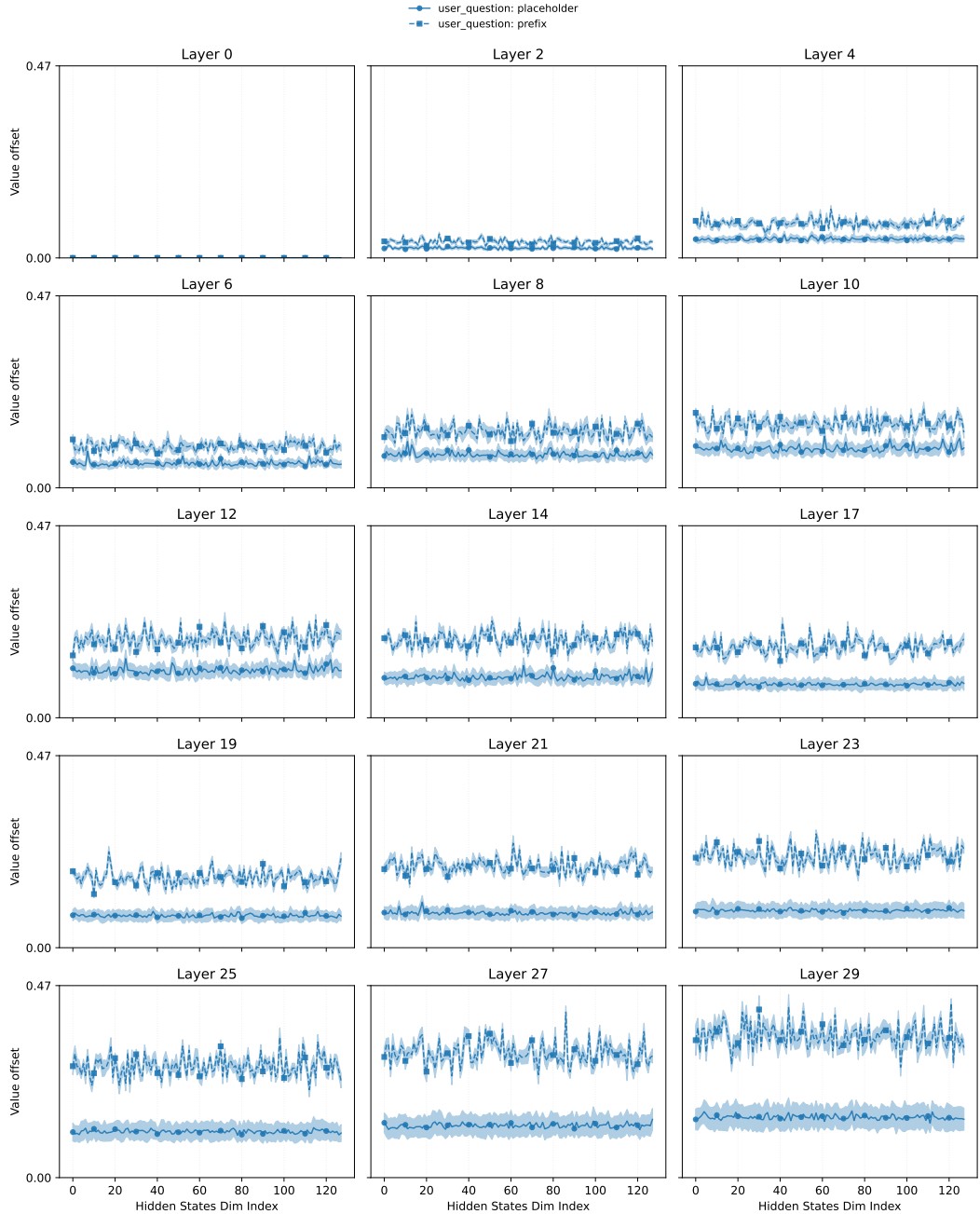

Figure A.5: Value cache offset distributions of the first agent's placeholder and prefix segments in a four-agent setting on the ten samples from the MMLU dataset.

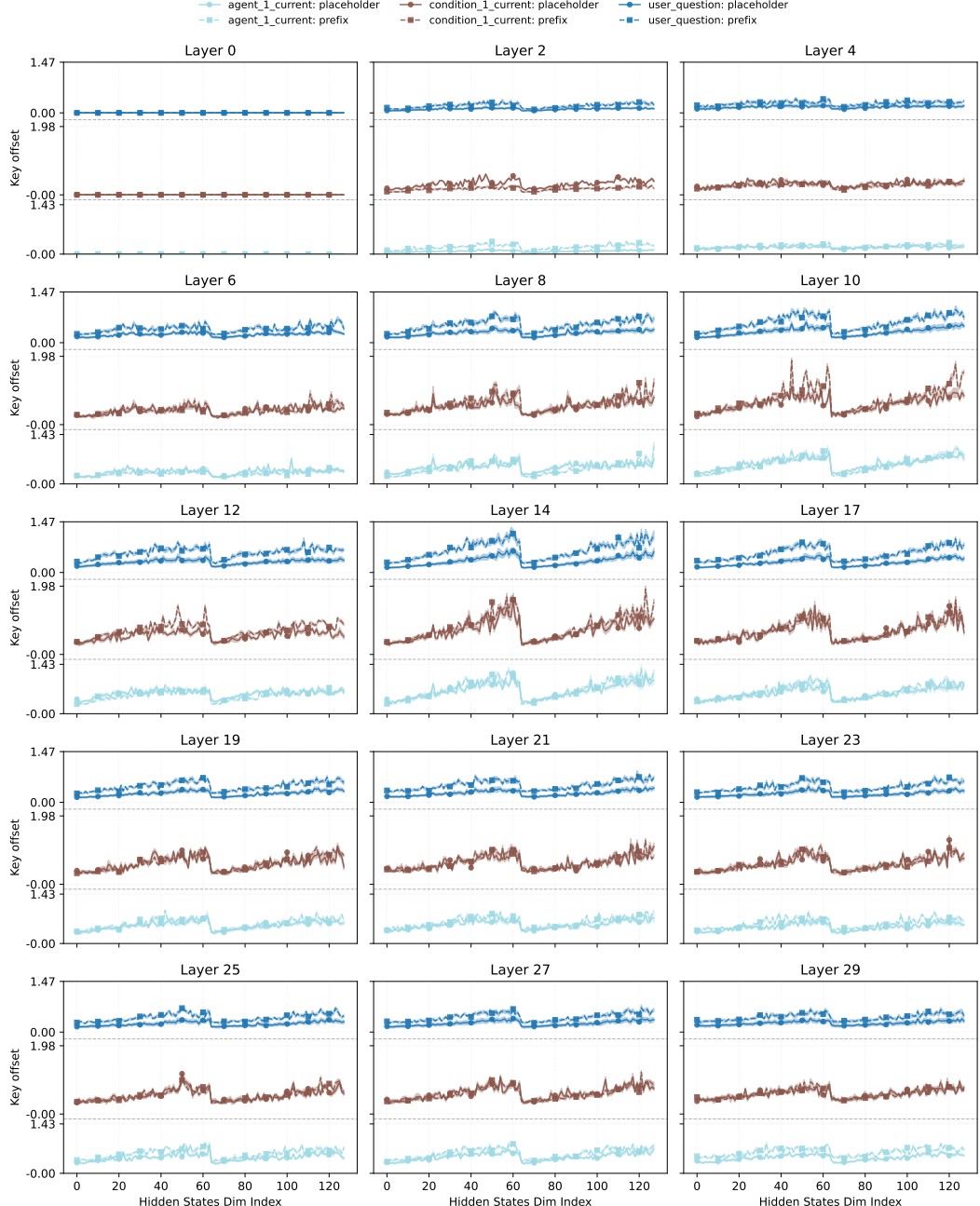

Figure A.6: Key cache offset distributions of the second agent's placeholder and prefix segments in a four-agent setting on the ten samples from the MMLU dataset.

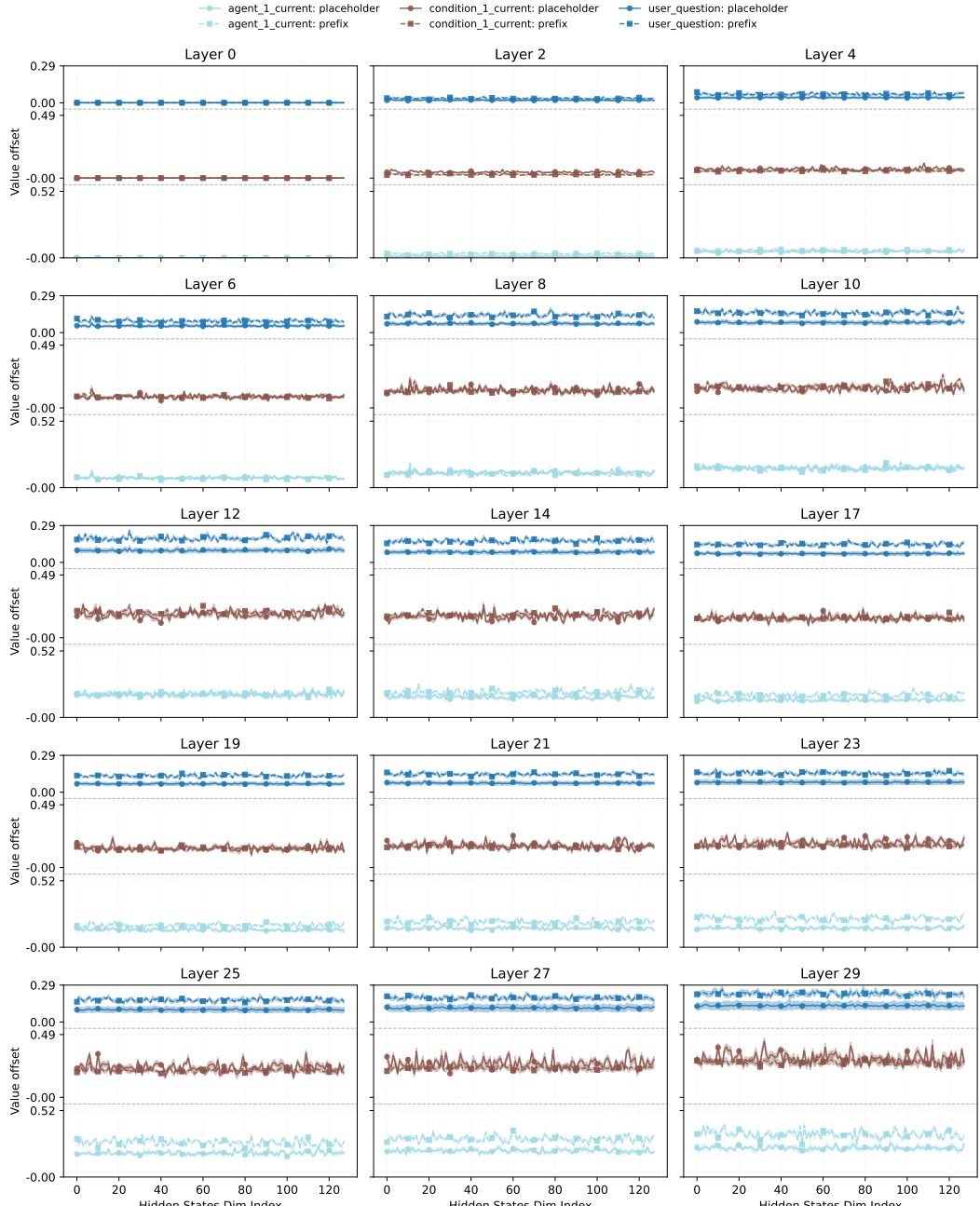

Figure A.7: Value cache offset distributions of the second agent's placeholder and prefix segments in a four-agent setting on the ten samples from the MMLU dataset.

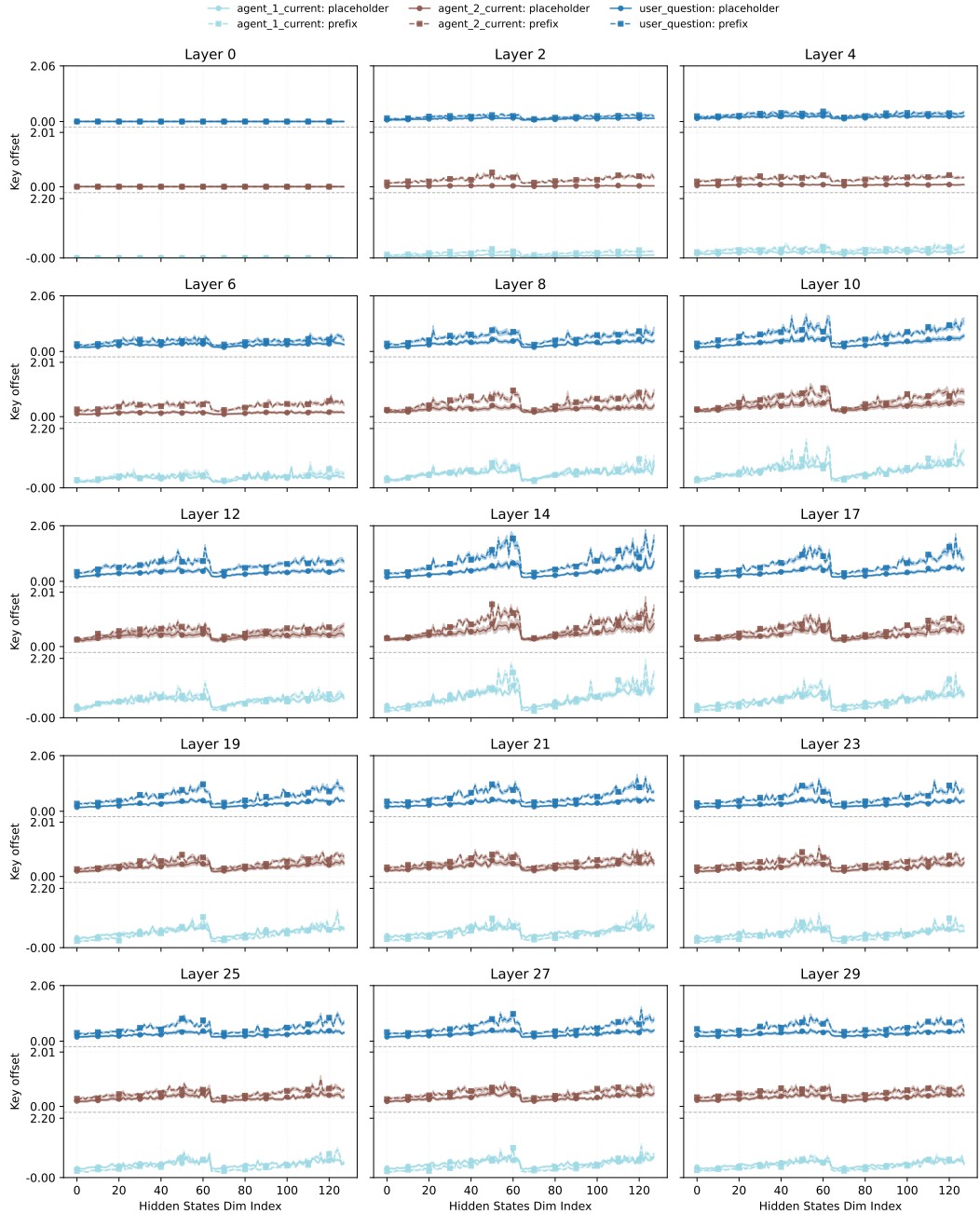

Figure A.8: Key cache offset distributions of the third agent's placeholder and prefix segments in a four-agent setting on the ten samples from the MMLU dataset.

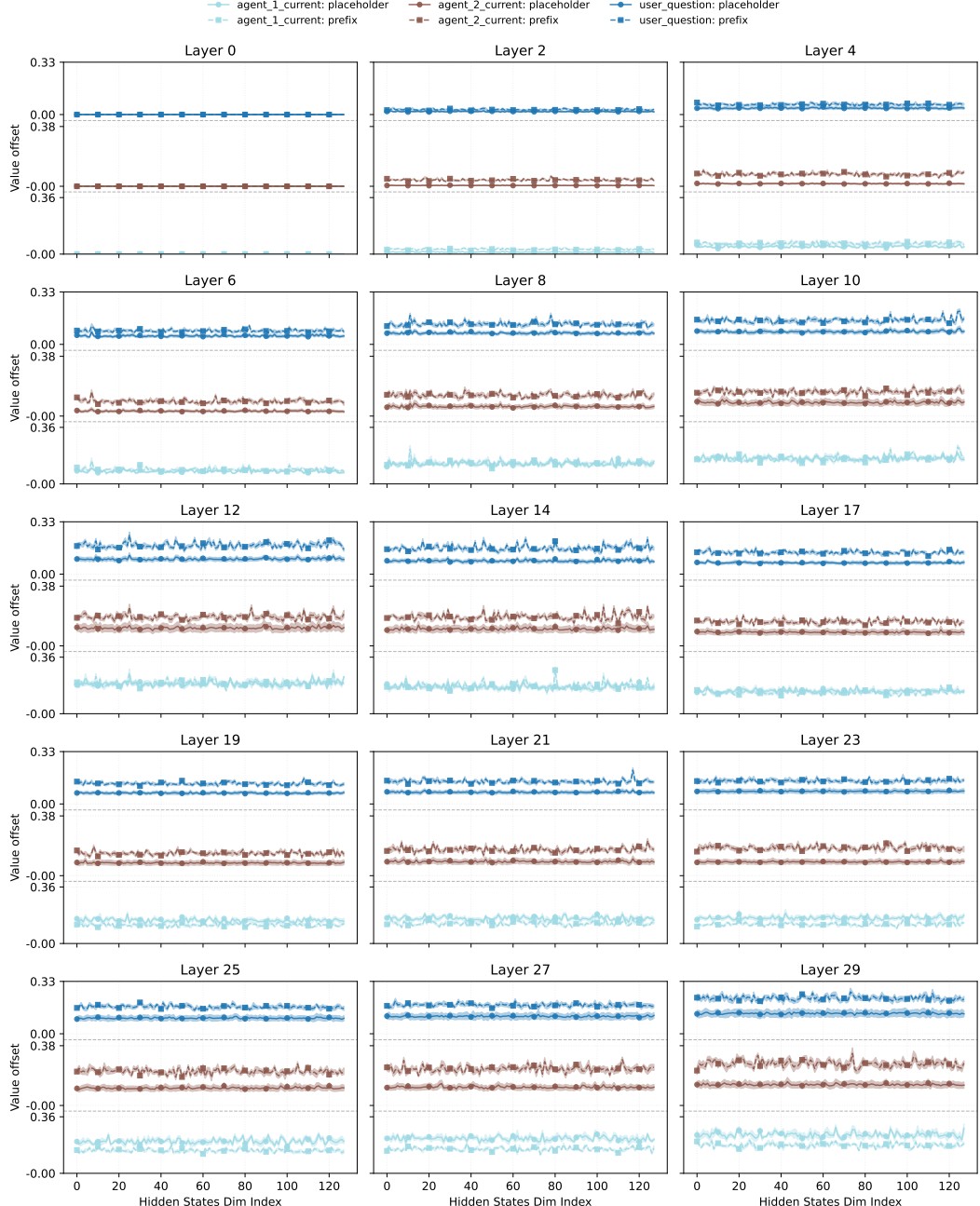

Figure A.9: Value cache offset distributions of the third agent's placeholder and prefix segments in a four-agent setting on the ten samples from the MMLU dataset.

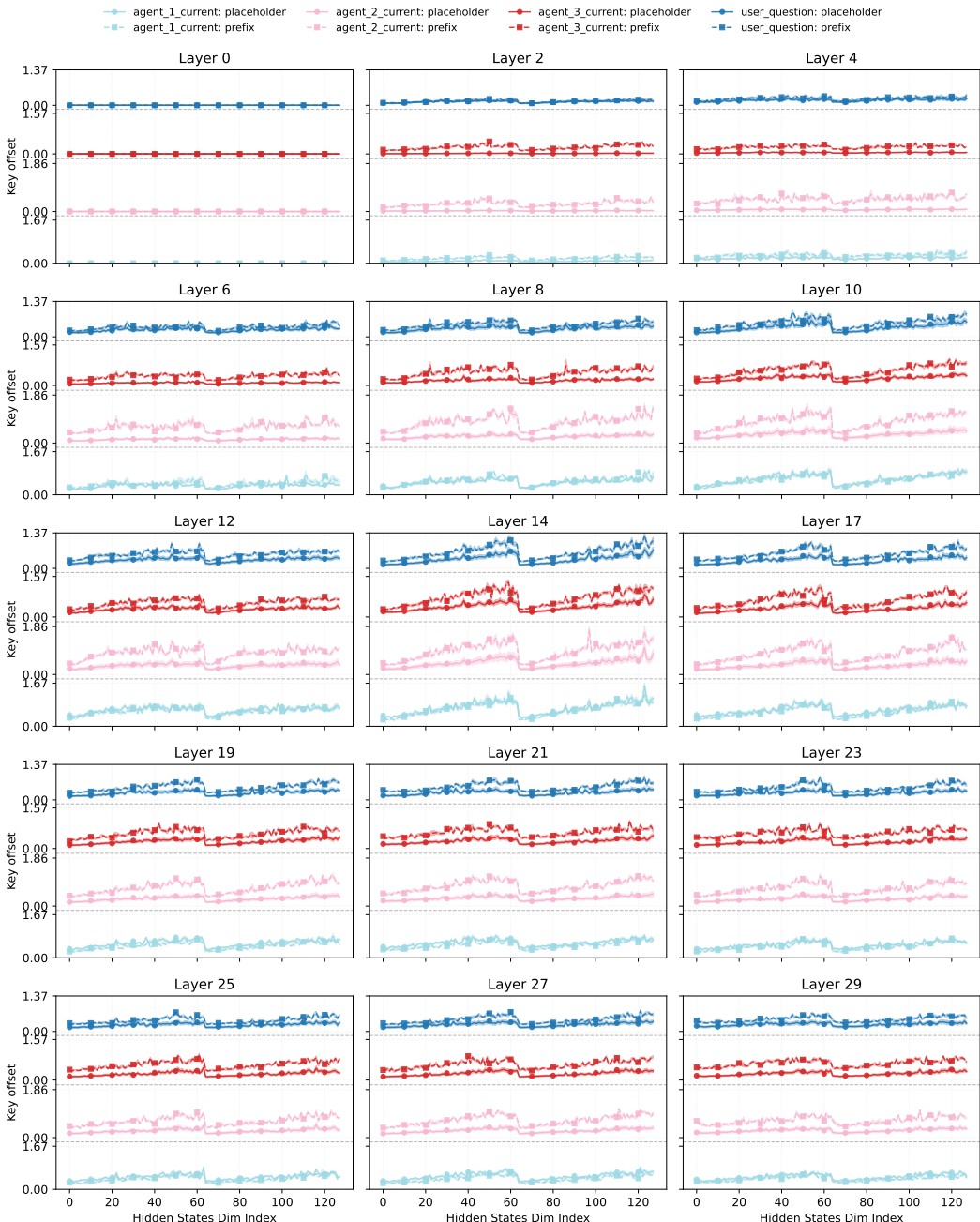

Figure A.10: Key cache offset distributions of the fourth agent's placeholder and prefix segments in a four-agent setting on the ten samples from the MMLU dataset.

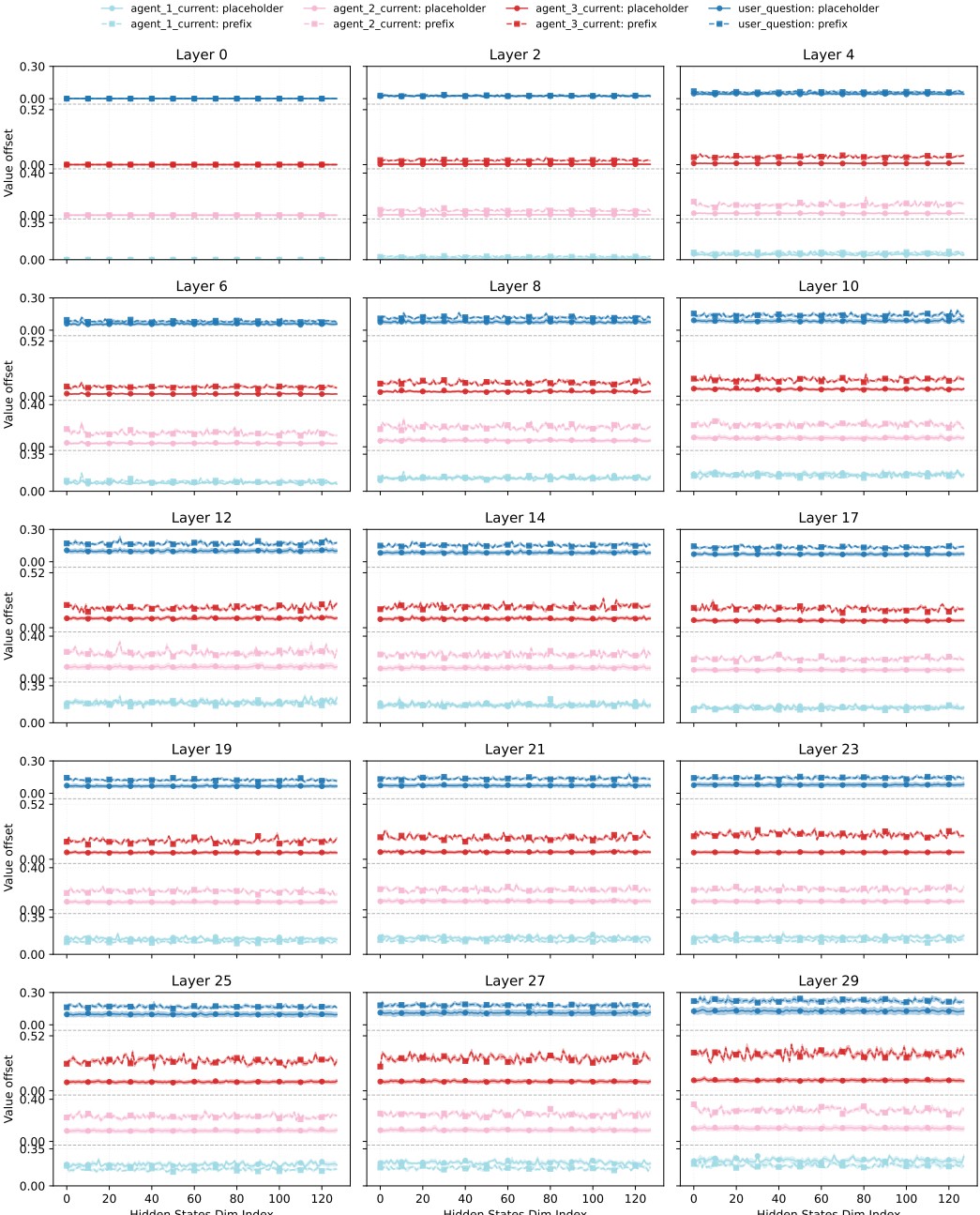

Figure A.11: Value cache offset distributions of the fourth agent's placeholder and prefix segments in a four-agent setting on the ten samples from the MMLU dataset.

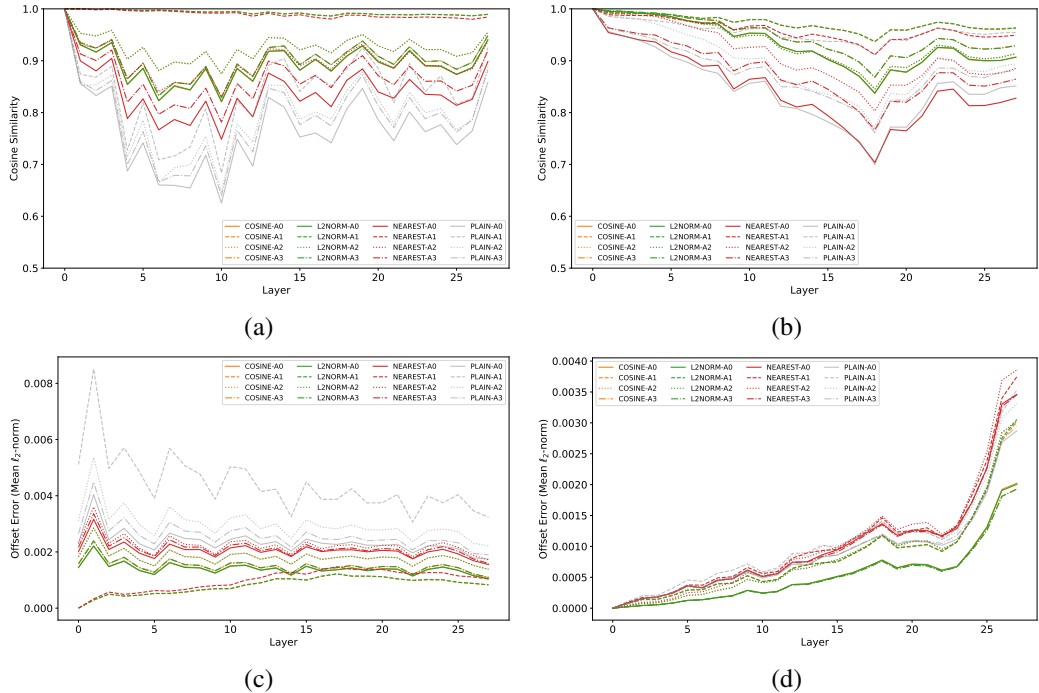

(a)                           (b)

(c)                           (d)

Figure A.12: Similarity and offset error analysis of approximated KV-caches versus real KV-caches across layers of Qwen-Coder-2.5-7B on HumanEval under a four-agent setting. (a) Cosine similarity distributions between approximated and real **key** caches. (b) Cosine similarity distributions between approximated and real **value** caches. (c) Mean $\ell_2$ norm error distributions between approximated and real **key** caches. (d) Mean $\ell_2$ norm error distributions between approximated and real **value** caches. Labels "COSINE-A0" to "COSINE-A3" denote cosine-similarity-based approximation; "L2NORM-A0" to "L2NORM-A3" denote our $\ell_2$-norm-based approximation; "NEAREST-A0" to "NEAREST-A3" indicate nearest-anchor sampling approximation; "PLAIN-A0" to "PLAIN-A3" represent unaligned baseline reuse.

consistently high cosine similarities (approximately 0.92 for keys and 0.95 for values) comparable to the cosine-based approach, while maintaining minimal offset errors across all layers. Unlike simpler methods such as nearest-reusing—which suffers substantial deviations in deeper layers (with the mean offset error exceeding 0.003 beyond layer 25)—our method robustly leverages weighted aggregation of multiple similar anchors to effectively estimate the target KV-caches. Additionally, the plain reuse baseline demonstrates severe similarity degradation (below 0.8 cosine similarity) and significantly elevated offset errors (above 0.004), confirming the critical importance of our fine-grained anchor alignment strategies, especially in deeper transformer layers where mismatch errors tend to accumulate.

