# OpenReview forum: "KVCOMM: Online Cross-context KV-cache Communication for Efficient LLM-based Multi-agent Systems"
_NeurIPS.cc/2025/Conference — NeurIPS 2025 poster_

### Official Review · Reviewer_y5Xb · 2025-06-23

**Clarity:** 3
**Significance:** 4
**Originality:** 4
**Rating:** 5
**Confidence:** 3

**Summary:**

This paper raises a key question: how can KV cache-based accelerated inference be effectively achieved in a multi-agent setting? It proposes a training-free calibration method, KVComm, which leverages KV cache offsets to enable kvcache reuse on placeholder texts with different prefixes.

**Questions:**

- What is the exact difference between the placeholder offset and the prefix offset? Could the author provide concrete examples to illustrate this? Why are both types of offsets necessary for multi-agent KV cache reuse?
- Is there any quantitative experimental result demonstrating the relationship between the number of anchors, GPU memory usage, and acceleration efficiency?

**Ethical Concerns:**

["NO or VERY MINOR ethics concerns only"]

**Final Justification:**

I increased the clarity score to 3 because the reviewer provided a detailed explanation of the overall process of KVCOMM in the rebuttal. I will keep my rating to 5 and advocate for acceptance.

**Limitations:**

Yes

**Paper Formatting Concerns:**

no concern.

**Quality:**

4

**Strengths And Weaknesses:**

Strengths:
- This paper provides both theoretical and empirical evidence for a compelling observation: despite variations in prefixes, identical placeholder texts often yield similar KV caches, and the discrepancy between them can be effectively captured and exploited to accelerate inference in multi-agent settings.
- It is the first KV cache framework specifically designed for multi-agent scenarios, offering valuable infrastructure for advancing research in the multi-agent domain.
- The proposed method achieves effective acceleration without requiring any additional training, making it lightweight and practical.

Weaknesses:
- The writing is somewhat hard to follow, and Figure 3 is overly complicated, making it difficult to grasp the overall process.
- The paper introduces too many technical terms, making it unclear how the proposed method is concretely implemented based on the intuitive idea of “KV cache offset.”

---

> ### Author Rebuttal · Authors · 2025-07-31
>
> Thank the reviewer for the time to review and provide the positive, constructive and valuable feedback. Below, we present our point-by-point responses to each comment.
>
> ## Q1: Overall Process in Figure 3
> Figure 3 is designed to illustrate the overall workflow of KVComm. We summarize the process as follows:
>
> 0. **Initialization**: Before any user requests, all agents precompute and store the KV-caches for all prefix segments defined in their prompt templates.
> 1. **Placeholder Readiness**: When a request arrives, each agent checks whether all placeholders’ base KV-caches are available. If not, these are precomputed in parallel. Newly generated placeholder KV-caches then enter the anchor prediction module to check for similar samples in the anchor pool and enable reuse.
> 2. **Reuse or Fallback**: Once all placeholder KV-caches are ready, the agent determines if there are reusable KV-cache deviations for each placeholder. If not, standard dense prefilling is used. For placeholders without reusable deviations, the agent computes the difference between their actual and base KV-caches, and stores this deviation in the anchor pool to expand anchor coverage.
> 3. **Offset Approximation**: If all placeholders have reusable KV-cache deviations, the agent fetches the matched anchors, estimates the KV-cache deviations using Eq. (6) and (7), and updates the placeholder and prefix KV-caches in parallel.
> 4. **Decoding**: The agent then combines the updated placeholder and prefix KV-caches to initiate response decoding.
> 5.	**Anchor Update**: After generating a response, its KV-cache passes through the anchor prediction module. If a similar anchor exists, the cache is stored in shared memory for future retrieval. The agent then awaits the next request.
> 6. **Fallback Storage**: If no similar anchor exists, the response KV-cache is stored in the anchor pool for dependent agents to fill in deviations under their respective contexts, then the agent awaits the next request.
>
> In conclude, **all agent interactions occur via the KV-cache Communication Module**. When multiple agents share the same user text but have different agent-specific prefixes, we avoid re-running prefilling by treating the KV-cache of the shared text under a new prefix as a context-dependent offset from its base KV-cache. We estimate that offset by interpolating from a small set of anchor examples, align Key positions (RoPE de-rotation / re-rotation), add the estimated Key/Value offsets, then concatenate the adjusted segments and decode.
>
> We will clarify and streamline Figure 3 in the revised manuscript to further improve its readability. If further clarification is needed, we are happy to provide it.
>
> ## Q2: Explanation of New Technical Terms
> We will add a glossary in the revised manuscript for readability and open-source our code for reproducibility. Below are the newly introduced terms in KVComm:
>
> 1. **Base KV-cache**: KV-cache for a prefix/placeholder under its initial context or without external input; serves as the reference for offsets.
>
> 2. **KV-cache offset/deviation**: The difference between a shared text’s KV-cache under a new prefix and its base KV-cache. (Keys require RoPE-based alignment before offsetting.)
>
> 3. **Placeholder/Prefix offset**: Offsets for the placeholder segment (external input) and its adjacent prefix segment (predefined context), respectively, relative to their base KV-caches.
>
> 4. **Offset variance problem**: KV-cache offsets for the same text can vary substantially across contexts, so static reuse is unreliable.
>
> 5. **Positional alignment/Key de-rotation**: RoPE de-rotation/re-rotation to align keys before offsetting.
>
> 6. **Shareability**: Whether a request can skip prefill under anchor criteria; if not, dense prefilling is used.
>
> 7. **Shared memory/KV-cache**: The shared storage that holds **base KV-caches** across agents.
>
> 8. **Dense prefilling/generation**: Full computation path (prefill + decode) used if reuse is not possible.
>
> 9. **Anchor (pool)**: A small set of representative examples, each storing placeholder and prefix offsets, used to interpolate offsets for new contexts.
>
>
> ## Q3: Deeper Discussion about the Placeholder and Prefix Offsets
> The exact difference between placeholder and prefix offset can be described from the following three aspects:
>
> 1. **Definition**:
>    - Placeholder offset is defined as the KV-cache offset for externally injected text, including users, agents, and tools, etc, whose base KV-cache is not preset;
>    - Prefix offset is defined as the KV-cache offset for predefined prompt text, including system prompt, placeholder conjunctions, and suffix.
>
> 2. **Dependence**:
>    - Placeholder offset arises mainly from changes to the prefix context;
>    - Prefix offset is triggered by changes to the preceding placeholder (i.e., injected content).
>
> 3. **Variance**:
>    - Placeholder offset show high variance across samples and contexts;
>    - Prefix offset are more stable, as their base KV-caches are computed under a fixed system prompt and are less influenced by preceding placeholders.
>
> Due to the rebuttal format restriction, we are currently unable to share visualization results of placeholder and prefix offset distribution in the response. We will add this interesting characteristics into the revised manuscript.
>
> **Concrete Example (from Figure 3)**
>
> Suppose the prompt template of an agent is defined as follows:
>
> *"You are a math expert. You will be given a math problem .... Q: {user_question}. Answers from other agents are: \n Agent 2 as a mathematical analyst: {agent_2_current}, the execution result is: {condition_2_current}.<|assistant|>"*
>
> In this case,
> - “You are a math expert…” is a prefix segment (fixed context).
> - “{user_question}”, “{agent_2_current}”,  {condition_2_current}” are placeholders (external content).
> - Text following each placeholder (e.g., “Answers from…”) is a prefix segment adjacent to a placeholder.
>
> From this example, we can find that during the precomputation of prefix KV-caches, the subsequent prefix segments are mainly correlated to the first prefix segment, typically containing system prompts, thus their base KV-caches are relatively stable. When the preceding placeholder is injected with specific text, their actual KV-cache will contain the new information. However, the base KV-cache of prefix segments, unaware of this, must be adjusted via offset interpolation using similar anchor cases. **Otherwise, the model may receive conflicting signals: the base KV-cache of prefix segments would mislead the model decode by showing no actual placeholder text injected, while the injected placeholder text actually participate in the information fusion. Therefore, this conflict necessitates careful offset estimation for both types.** Experimental demonstration of the necessity has been provided in Table 5 of the main text.
>
> Based on the above analysis, hopefully, it would be clearer about the difference between the two offset types and the necessity of them. If further clarification is needed, we are happy to provide it.
>
> ## Q4: Relationship between the number of anchors, the GPU memory usage, and TTFT speedup
> We provide detailed experimental results in Tables 1 and 2, and analyze the relationship as follows:
>
> **Memory Cost vs. Anchor Number:**
>
> As shown in Table 1, GPU memory usage increases approximately linearly with the number of anchors, for fixed input/output lengths and agent count. This trend holds across all sequence lengths: each additional anchor increases storage for associated KV deviations, leading to higher overall memory cost.
>
> **TTFT Speedup vs. Anchor Number:**
>
> However, in Table 2, the TTFT speedup does not increase monotonically with more anchors. Specifically,
> when increasing anchor number from 5 to 10 or 15, the TTFT speedup remains nearly unchanged or even slightly decreases across most settings. This indicates that anchor pool enlargement yields diminishing returns, and excessive anchors may even cause minor efficiency loss due to increased system overhead (e.g., lookup, memory access, or less optimal anchor matching).
>
> Table 1. Memory Cost (GB) under the three-agent setting with different maximun anchor numbers.
>
> |IN|OUT|#anchor=5|#anchor=10|#anchor=15|
> |-|-|-|-|-|
> |64|128|1.67|3.31|4.95|
> |64|256|1.98|3.94|5.88|
> |64|512|2.61|5.19|7.76|
> |64|1024|3.86|7.69|11.5|
> |128|128|1.70|3.33|4.97|
> |128|256|2.01|3.96|5.91|
> |128|512|2.64|5.21|7.78|
> |128|1024|3.89|7.71|11.5|
> |256|128|1.74|3.38|5.02|
> |256|256|2.06|4.01|5.95|
> |256|512|2.68|5.26|7.83|
> |256|1024|3.93|7.76|11.6|
> |512|128|1.84|3.47|5.11|
> |512|256|2.15|4.10|6.05|
> |512|512|2.78|5.35|7.92|
> |512|1024|4.02|7.85|11.7|
> |1024|128|2.03|3.66|5.30|
> |1024|256|2.34|4.29|6.23|
> |1024|512|2.96|5.54|8.11|
> |1024|1024|4.21|8.04|11.9|
>
> Table 2. Mean TTFT speedup under the three-agent setting with different maximun anchor numbers.
>
> |IN|OUT|#anchor=5|#anchor=10|#anchor=15|
> |-|-|-|-|-|
> |64|128|5.24|5.17|4.41|
> |64|256|5.64|5.82|4.86|
> |64|512|6.80|6.76|6.10|
> |64|1024|8.72|8.92|8.19|
> |128|128|5.50|5.40|4.77|
> |128|256|5.60|5.55|5.35|
> |128|512|6.91|6.94|6.67|
> |128|1024|9.01|9.09|8.55|
> |256|128|5.99|5.92|5.00|
> |256|256|6.15|6.12|5.52|
> |256|512|7.37|7.62|6.91|
> |256|1024|9.22|9.53|8.59|
> |512|128|7.02|6.98|5.89|
> |512|256|7.65|7.64|4.70|
> |512|512|8.77|8.65|8.19|
> |512|1024|10.38|10.52|10.11|
> |1024|128|9.97|9.66|8.56|
> |1024|256|10.18|10.42|9.05|
> |1024|512|10.80|11.09|10.55|
> |1024|1024|12.79|12.94|12.32|

---

> > ### Comment · Reviewer_y5Xb · 2025-08-03
> >
> > I increased the clarity score to 3 because the reviewer provided a detailed explanation of the overall process of KVCOMM in the rebuttal. I suggest adding this explanation to the final version of the paper. I will keep my rating to 5 and advocate for acceptance.

---

> > > ### Author Response · Authors · 2025-08-03
> > >
> > > We greatly appreciate the reviewer’s positive feedback and sincerely thank the reviewer for advocating for acceptance. We will incorporate the detailed explanation of the KVComm process and supplementary results into the final manuscript, and open-source the code repository that integrates KVComm with modern LLM serving frameworks to facilitate the deployment of efficient multi-agent systems.

---

### Official Review · Reviewer_3BgL · 2025-06-29

**Clarity:** 3
**Significance:** 3
**Originality:** 3
**Rating:** 4
**Confidence:** 3

**Summary:**

The authors identify the offset-variance problem that prevents naïve KV-cache sharing across agents, and propose KVCOMM—a training-free, anchor-based framework. The anchor pool will store representative anchors on the fly, interpolates and learns the offsets across layers to align future shared tokens, and reuses the saved anchors to accelerate the prefilling. KVCOMM can achieve up to 13.7 speedup with minimal performance degradations.

**Questions:**

In the Table1, the performance of original settings on MMLU is quite low (47.1). Is it a typo?

**Ethical Concerns:**

["NO or VERY MINOR ethics concerns only"]

**Final Justification:**

In the review, authors provided detailed experiments on the breakdown of the overhead, comparison over different baselines, and experiments on harder tasks. I will raise my score by 1 since authors give detailed experimentation to resolve my concerns, yet I still notice the E2E improvement is not very exciting.

**Limitations:**

My main concerns are in the evaluation sections. It would be better if the author can provide more quantitative results. Please see my comments in the weakness section.

**Paper Formatting Concerns:**

Not included

**Quality:**

2

**Strengths And Weaknesses:**

- **Strengths**
  - Tackles the important problem of improving the efficiencies of multi-agent systems (MAS).
  - Grounded with solid theory, providing formal error bounds and clear motivation.
  - Reframes KV reuse as an *offset-alignment* problem and proposes an anchor-based interpolation mechanism.
  - Includes extensive hyper-parameter ablation studies demonstrating the method’s robustness.

- **Weaknesses**
  - Experiments cover only a limited set of tasks and MAS setups; harder benchmarks such as Math500, AIME, and GAIA are not evaluated.
  - Empirical comparison focuses mainly on a single baseline (cache blend). More possible baselines are helpful to get a comprehensive evaluation
  - End-to-end (E2E) latency is not reported. It is unclear how much the speed-up of TTFT can be transferred to the E2E latencies and throughputs gains, which may be concerned more in the MAS settings.
  - The anchor pools may introduce signifcant overhead, but this cost is not quantified. The KV cache could be very large for large models and complicated MAS trajectories.
  - More fine-grained breakdowns are needed—for example, measuring the time to load retrieved elements into GPU memory when anchor pools are large and cannot be fully stored on-GPU.
  - Sharing caches could raise safety or privacy concerns. Although this has already been discussed in the paper, no concrete measures is proposed to solve this concerns.

---

> ### Author Rebuttal · Authors · 2025-07-31
>
> Thank the reviewer for the time to review and provide the valuable and constructive feedback. We address each of the comments in detail below.
> ## Q1: Evaluations on Harder Benchmarks
>
> For Math500, we report results using both Llama3.1-8B-instruct (Llama) and Deepseek-R1-Distill-Qwen-7B (DS-QWen). As shown in Tables 1 and 2, **KVComm achieves competitive or superior accuracy compared to the baseline method**. Notably, even as reuse rate drops with increasing agent number (due to higher context diversity), KVComm consistently matches or exceeds the baseline accuracy, suggesting that referencing anchor information helps improve answer quality. An especially encouraging observation is that **for DS-QWen, KVComm attains both a higher reuse rate and superior accuracy on challenging reasoning tasks**.
>
> Table 1. Math500 Results – Llama.
>
> |Method|Metric|#Agent=2|#Agent=3|#Agent=4|
> |-|-|-|-|-|
> |Original|Accuracy|**41.6**|**39.6**|42.6|
> |KVComm|Accuracy|38.0|38.6|**44.2**|
> ||Reuse rate|59.4|40.9|30.7|
>
> Table 2. Math500 Results – DS-QWen.
>
> |Method|Metric|#Agent=2|#Agent=3|#Agent=4|
> |-|-|-|-|-|
> |Original|Accuracy|**51.4**|49.6|42.8|
> |KVComm|Accuracy|50.8|**50.8**|**45.8**|
> ||Reuse rate|76.7|60.4|45.3|
>
> For AIME, Table 3 reports the results using DS-QWen. **KVComm maintains a high reuse rate (70%+) while retaining comparable accuracy to the baseline**. The accuracy decrease largely attributes to token-length constraints during decode: as KVComm’s memory cost rises, the available context for generation per agent is reduced. We are actively addressing this with pipelining and efficient offloading techniques to support longer contexts in future work.
>
> Table 3. Performance on AIME using DS-QWen models.
>
> |Method|Metric|#Agent=2|#Agent=3|#Agent=4|
> |-|-|-|-|-|
> |Original|Accuracy|**19.2**|**17.5**|**17.5**|
> |KVComm|Accuracy|11.7|10.8|8.3|
> ||Reuse rate|71.3|78.1|74.6|
> ## Q2: Comparison with More Methods
>
> Below we report experimental results for DroidSpeak alongside KVComm. DroidSpeak is designed for scenarios where **the base and finetuned models process the same entire prompt**—thus, context change is not addressed.
>
> Table 4 demonstrates that applying DroidSpeak directly in our experimental setup significantly degrades performance, **validating the necessity of context-aware cache reuse. KVComm consistently achieves higher or comparable accuracy across all datasets**.
>
> Table 4. Performance comparison with DroidSpeak.
>
> |Dataset|Method|#Agent=2|#Agent=3|#Agent=4|
> |-|-|-|-|-|
> |MMLU|Original|47.1|66.7|68.0|
> ||DroidSpeak|30.1|40.5|38.6|
> ||KVComm|**64.7**|**68.6**|**68.0**|
> |GSM8K|Original|81.1|**82.4**|**82.1**|
> ||DroidSpeak|52.2|20.6|14.5|
> ||KVComm|**81.5**|81.7|80.6|
> |HumanEval|Original|**86.3**|**83.9**|**84.5**|
> ||DroidSpeak|14.9|27.3|15.5|
> ||KVComm|81.4|83.2|83.2|
> |Math500|Original|**41.6**|**39.6**|42.6|
> ||DroidSpeak|14.8|14.8|14.8|
> ||KVComm|38.0|38.6|**44.2**|
> ## Q3: End-to-End (E2E) Latency Report
> We would like to clarify that in the KV-cache reuse literature (e.g., PromptCache, CacheBlend, DroidSpeak, KVLink), most works do not specifically target E2E latency optimization because E2E latency encompasses both prefill and decoding, whereas KV-cache reuse specifically accelerates prefill (TTFT). However, it has been shown (e.g., PromptCache) that in modern serving frameworks supporting prefill/decoding disaggregation (e.g., DistServe, Dynamo), reducing TTFT can significantly improve system throughput and user-perceived latency.
>
> Nevertheless, for completeness, we provide E2E latency for various agent settings (Table 5). While the E2E latency improvement is modest (up to $\sim$ 1.08x in long-prefill/short-decode cases), this is expected since KVComm does not target the decode stage. Still, any E2E improvement directly reflects the benefit of reduced prefill.
>
> Table 5. Latency Breakdown and Speedup vs. Dense Prefill.
>
> |IN|OUT|Agent ID|KVComm (s)|Other (s)|TTFT_Ori (s)|Decoding (s)|TTFT Speedup|E2E Speedup|
> |-|-|-|-|-|-|-|-|-|
> |64|128|0|0.083|0.005|0.068|2.430|0.770|0.992|
> |64|128|1|0.007|0.006|0.080|2.435|6.017|1.027|
> |64|128|2|0.008|0.008|0.097|2.434|6.088|1.033|
> |64|256|0|0.090|0.005|0.089|4.911|0.930|0.999|
> |64|256|1|0.008|0.007|0.115|4.919|8.199|1.020|
> |64|256|2|0.008|0.008|0.144|4.913|8.667|1.026|
> |64|512|0|0.090|0.005|0.089|9.913|0.934|0.999|
> |64|512|1|0.008|0.007|0.139|9.903|8.924|1.012|
> |64|512|2|0.009|0.010|0.213|9.910|11.130|1.020|
> |64|1024|0|0.090|0.005|0.087|19.997|0.913|1.000|
> |64|1024|1|0.010|0.009|0.204|20.007|10.762|1.009|
> |64|1024|2|0.011|0.013|0.369|20.004|15.260|1.017|
> |128|128|0|0.090|0.005|0.094|2.492|0.986|0.999|
> |128|128|1|0.007|0.006|0.110|2.499|8.203|1.038|
> |128|128|2|0.008|0.008|0.122|2.499|7.985|1.043|
> |128|256|0|0.090|0.005|0.093|4.928|0.982|1.000|
> |128|256|1|0.009|0.007|0.120|4.929|7.901|1.021|
> |128|256|2|0.008|0.008|0.149|4.927|8.987|1.027|
> |128|512|0|0.090|0.005|0.094|9.946|0.981|1.000|
> |128|512|1|0.009|0.007|0.145|9.956|9.084|1.013|
> |128|512|2|0.009|0.010|0.220|9.967|11.299|1.020|
> |128|1024|0|0.091|0.005|0.092|19.945|0.958|1.000|
> |128|1024|1|0.010|0.009|0.211|19.955|10.947|1.010|
> |128|1024|2|0.011|0.013|0.377|19.945|15.372|1.018|
> |256|128|0|0.083|0.005|0.082|2.461|0.932|0.998|
> |256|128|1|0.007|0.006|0.098|2.466|7.092|1.034|
> |256|128|2|0.008|0.008|0.115|2.467|7.281|1.040|
> |256|256|0|0.090|0.005|0.107|4.947|1.125|1.002|
> |256|256|1|0.009|0.007|0.134|4.943|8.716|1.024|
> |256|256|2|0.008|0.009|0.162|4.950|9.561|1.029|
> |256|512|0|0.090|0.005|0.108|9.891|1.129|1.001|
> |256|512|1|0.009|0.007|0.159|9.891|9.769|1.014|
> |256|512|2|0.009|0.010|0.239|9.893|12.234|1.022|
> |256|1024|0|0.092|0.005|0.106|20.190|1.091|1.000|
> |256|1024|1|0.011|0.009|0.230|20.211|11.514|1.010|
> |256|1024|2|0.012|0.014|0.399|20.160|15.698|1.018|
> |512|128|0|0.091|0.006|0.132|2.473|1.366|1.014|
> |512|128|1|0.007|0.006|0.144|2.474|10.366|1.052|
> |512|128|2|0.008|0.008|0.162|2.474|10.181|1.059|
> |512|256|0|0.090|0.006|0.131|4.980|1.377|1.007|
> |512|256|1|0.008|0.007|0.159|4.992|10.620|1.029|
> |512|256|2|0.009|0.009|0.206|4.983|11.621|1.038|
> |512|512|0|0.090|0.006|0.132|9.950|1.377|1.004|
> |512|512|1|0.009|0.007|0.202|9.955|12.329|1.019|
> |512|512|2|0.010|0.010|0.272|9.952|13.558|1.025|
> |512|1024|0|0.093|0.006|0.130|20.041|1.324|1.002|
> |512|1024|1|0.011|0.010|0.264|20.006|12.898|1.012|
> |512|1024|2|0.011|0.014|0.441|19.971|17.264|1.021|
> |1024|128|0|0.091|0.006|0.200|2.501|2.055|1.039|
> |1024|128|1|0.008|0.007|0.217|2.510|14.053|1.080|
> |1024|128|2|0.009|0.009|0.237|2.506|13.628|1.087|
> |1024|256|0|0.092|0.006|0.200|4.972|2.052|1.020|
> |1024|256|1|0.009|0.007|0.236|4.987|14.630|1.044|
> |1024|256|2|0.009|0.009|0.271|4.987|14.401|1.050|
> |1024|512|0|0.092|0.006|0.200|9.869|2.045|1.010|
> |1024|512|1|0.010|0.008|0.266|9.881|14.499|1.025|
> |1024|512|2|0.011|0.011|0.361|9.890|16.919|1.034|
> |1024|1024|0|0.092|0.006|0.195|20.055|1.992|1.005|
> |1024|1024|1|0.011|0.009|0.344|20.027|16.459|1.016|
> |1024|1024|2|0.013|0.014|0.550|19.988|20.565|1.026|
> ## Q4: Memory Overhead Report Induced by KVComm
> Below we provide the memory footprint for different agentic settings, including varying configurations of input/output length, and anchor numbers. As shown in Table 6, memory usage scales with both sequence and anchor number. Note that it is observed that **the deviation tensors stored for each anchor are highly sparse (averaging ~50% elements with absolute value < 1e-1), suggesting potential for lossless compression of anchors and further memory optimization**.
>
> Table 6. Memory Cost (GB) under the three-agent setting.
>
> |IN|OUT|#anchor=5|#anchor=10|#anchor=15|#anchor=20|
> |-|-|-|-|-|-|
> |64|128|1.67|3.31|4.95|6.58|
> |64|256|1.98|3.94|5.88|7.83|
> |64|512|2.61|5.19|7.76|10.3|
> |64|1024|3.86|7.69|11.5|15.3|
> |128|128|1.70|3.33|4.97|6.60|
> |128|256|2.01|3.96|5.91|7.85|
> |128|512|2.64|5.21|7.78|10.4|
> |128|1024|3.89|7.71|11.5|15.4|
> |256|128|1.74|3.38|5.02|6.65|
> |256|256|2.06|4.01|5.95|7.90|
> |256|512|2.68|5.26|7.83|10.4|
> |256|1024|3.93|7.76|11.6|15.4|
> |512|128|1.84|3.47|5.11|6.75|
> |512|256|2.15|4.10|6.05|8.00|
> |512|512|2.78|5.35|7.92|10.5|
> |512|1024|4.02|7.85|11.7|15.5|
> |1024|128|2.03|3.66|5.30|6.9|
> |1024|256|2.34|4.29|6.23|8.18|
> |1024|512|2.96|5.54|8.11|10.7|
> |1024|1024|4.21|8.04|11.9|15.7|
> ## Q5: Finegrained Breakdown for Large Anchor Scenarios
> Table 2 in the main text has provided latency breakdowns and speedup, where the anchor loading overhead is counted at the 'Others' part in Table 2. For the large anchor scenario, we take the MMLU benchmark as an example with 4K placeholder tokens. Table 7 shows that anchor offloading can become a major latency bottleneck (up to 1.3 seconds) and that cumulative CPU memory can reach 75GB. While this is significant, *it can be mitigated by engineering improvements such as asynchronous scheduling and anchor compression, which are the focus of our ongoing system development.*
>
> Table 7. 4K-token Anchor Matching using Llama on MMLU
>
> |Metrics|#Agent=3|#Agent=4|
> |-|-|-|
> |Softmax (ms)|104|122|
> |Offloading (ms)|1260|1300|
> |Accuracy (%)|66.7|68.0|
> |Reuse Rate (%)|49.7|51.0|
> |Memory Cost (GB)|48.5|75.1|
> ## Q6: Safety Considerations in Cache Sharing
> We have reviewed our submission and confirm that safety and privacy of KV-cache sharing are not discussed in this version. While we agree that safety and privacy are of critical importance in practical multi-agent systems, a comprehensive treatment is outside the current paper’s scope. We will highlight this as a promising and important direction for future work.
> ## Q7: Baseline Peroformance on MMLU
> We have double-checked our experiments and confirm the correctness of the reported accuracy for the baseline on MMLU. The lower baseline performance under the two-agent setting arises from our agent design: the first agent is a knowledgeable expert who generates relevant keywords, and the second "Final Refer" is tasked to analyze the previous agent’s output and produce a final answer. Some failures are due to the second agent providing only an answer without analysis. All prompt templates are available in Appendix 6.3.2.

---

> > ### Comment · Reviewer_3BgL · 2025-08-04
> >
> > Thanks for the discussion. This has resolved my concerns for lack of evaluations on different bencharks and different settings. I hope these experiments as well as the breakdown of the overhead can be added to the camera-ready version. I will raise my score.

---

> > > ### Author Response · Authors · 2025-08-04
> > >
> > > We greatly appreciate the reviewer's positive feedback and sincerely thank the reviewer for increasing the score. We will supplement all experimental results and analyses in the camera-ready version of the manuscript, and open-source the code repository that integrates KVComm with modern LLM serving frameworks to facilitate the deployment of efficient multi-agent systems.

---

### Official Review · Reviewer_TNJW · 2025-06-30

**Clarity:** 3
**Significance:** 3
**Originality:** 3
**Rating:** 4
**Confidence:** 4

**Summary:**

This paper addresses inefficient redundant KV-cache computation in multi-agent LLM systems caused by offset variance, where agents sharing overlapping contexts reprocess identical content due to divergent prefix lengths and dynamic placeholders, preventing static KV-cache reuse. The authors propose KVComm, a training-free framework enabling dynamic KV-cache reuse. KVComm employs an online anchor pool that tracks cache deviations under varying prefixes to estimate and align cache offsets for new contexts.

**Questions:**

1.Proposition 1 assumes RoPE-based positional encoding. How does KVComm handle non-RoPE models? Experiments on such architectures would strengthen claims.

2.The prefix segmentation requires rigid templates (Eq.1), making free-form agent interactions (e.g., unstructured debates) unsupported. The framework fails when placeholders aren't predefined—shown by omitting conversational benchmarks.

3.Tests use ≤1K tokens (Table 3). How does reuse rate degrade for >4K contexts where cumulative positional deviations amplify?

**Ethical Concerns:**

["NO or VERY MINOR ethics concerns only"]

**Final Justification:**

Most concerns are resolved. I maintain my original positive score.

**Limitations:**

If the method only applies to strictly identical LLMs and rigidly fixed prefixes, and cannot accommodate free-form conversational interactions, the authors may wish to explicitly acknowledge these constraints in the limitations section.

**Quality:**

3

**Strengths And Weaknesses:**

Strengths: The authors effectively combine theoretical analysis (Propositions 1 and 2) with extensive experiments across diverse benchmarks (MMLU, GSM8K, HumanEval). They demonstrate significant results, including a 13.7× TTFT speedup and over 70% reuse rate under realistic multi-agent workloads. Ablation studies (Table 5) validate the necessity of each component. The work further includes comprehensive robustness evaluations, assessing sensitivity to hyperparameters (Table 6), request ordering (Table 4), and context length (Table 3).

weakness:
1.The paper assumes identical LLMs for all agents, but real-world multi-agent systems often leverage heterogeneous models (e.g., domain-specific LLMs). This severely limits practical applicability.
2.Anchor matching relies on real-time embedding similarity calculations (Eq.5), yet latency overhead for long-context inputs is unmeasured. Given the softmax over anchors, this could negate prefilling speedup for large V. Please quantify this cost.
3.While DroidSpeak solves cross-model KV-sharing, it's only cited without comparison. This omission undermines the claimed novelty.

---

> ### Author Rebuttal · Authors · 2025-07-31
>
> Thank the reviewer for taking the time to review our work and providing positive, constructive and valuable feedback. Below, we present our point-by-point responses to each comment.
>
> ## Q1: Application in Heterogenous Multi-agent Systems
>
> As for the heterogenous systems, we can categorize them into two types for detailed discussion.
>
> For agents consisting of
> - **Identical Architecuture but Different Model Weights:** DroidSpeak has discovered the KV-cache reuse potential for this kind of KV-cache communication, where the base model and domain-specific finetuned model completely share the same prompt. Building on this, we believe that if the clustering characteristics of tokens in the finetuned model remain sufficiently similar to the base model, KVComm could facilitate KV-cache sharing for these cases. However, this may require further technical innovation and thus remains a promising direction for future research.
> - **Distinct Architectures**: In this case, it is impossible to employ KVComm as the reviewer noted.
>
> In summary, KVComm is **directly applicable for groups of homogeneous agents and holds promise for agents with identical architectures but different weights**, pending further exploration.
>
> ## Q2: Softmax Overhead in Long-context Anchor matching
>
> In KVComm, the softmax is operated along the anchor number dimension on the negative $\ell_2$-norm distance between the placeholder sample and each anchor, which reduces the Key/Value tensor into the shape of [$m$, 1, 1, seq_len, 1] ($m$: anchor count, seq_len: sequence length).  The latency of softmax thus scales with both parameters. And we quantify its latency overhead in Table 1. Here the shape of each KV-cache tensor is [32, 8, seq_len, 128]. We can observe that latency remains reasonable ($\sim$ 18 ms with 25 anchors and 4096 tokens per anchor).
>
> Table 1. Simulated softmax latency (ms) with different anchor counts and sequence length.
>
> |#Anchor|1024|2048|4096|
> |-|-|-|-|
> |5|0.894|1.719|3.552|
> |10|1.773|3.576|7.128|
> |15|2.620|5.332|10.766|
> |20|3.933|7.859|15.624|
> |25|4.435|9.614|18.113|
>
> These results are from a simulation without competing system workloads. In real multi-agent long-context scenarios, the latency also includes offloading KV-caches to CPU. For example, as shown in Table 2 (using Llama3.1-8B-instruct on the MMLU benchmark), the average softmax latency is 100+ ms when all anchors are offloaded to CPU, while total offloading per agent can reach 1260+ ms for 4K-token contexts. This indicates that **the main overhead arises from data movement, not from the softmax operation itself**. Such communication overhead is a well-known issue in long-context inference, and **can be mitigated with systematic optimization (e.g., pipelining), which is orthogonal to the KVComm mechanism**. To achieve perfect acceleration of long-context KV-cache reuse, *we are currently committed to integrate KVComm into the modern LLM serving framework such as vLLM and SGLang, which have well optimized these systematic issues yet still lack support for fine-grained, per-layer KV-cache injection across devices*.
>
> Table 2. Latency and Memory Cost of 4K-tokens Anchor Matching using Llama3.1-8B-instruct on the MMLU benchmark.
>
> |Metrics|#Agent=3|#Agent=4|
> |-|-|-|
> |Softmax (ms)|104|122|
> |Offloading (ms)|1260|1300|
> |Accuracy (%)|66.7|68.0|
> |Reuse Rate (%)|49.7|51.0|
> |Memory Cost (GB)|68.5|95.1|
>
> ## Q3: Experimental Comparisons with DroidSpeak
>
> The reason we did not originally compare directly to DroidSpeak is due to fundamental differences in application scenario:
>
> - **DroidSpeak** exploits cross-model KV-cache reuse when both base and finetuned models process **an identical prompt, so context changes are not involved**.
> - **KVComm** targets at the potential of cross-model KV-cache reuse **when only a shared segment is reused under different prefix contexts—a more general and challenging setting for multi-agent systems.**
>
> Therefore, although both designed based on KV-cache reuse, the two methods aim at solving two different problems in multi-agent systems.
>
> To clarify, we implemented DroidSpeak in our framework and empirically compared it against KVComm, holding DroidSpeak’s reuse rate at 80% for relative fairness. It can be observed in Table 3 that DroidSpeak’s performance drops significantly in our setting, confirming that **context awareness is crucial for robust KV-cache reuse**.
>
> Table 3. Performance comparison with DroidSpeak on various benchmarks using Llama3.1-8B-instruct.
>
> |Dataset|Method|#Agent=2|#Agent=3|#Agent=4|
> |-|-|-|-|-|
> |MMLU|Original|47.1|66.7|68.0|
> ||DroidSpeak|30.1|40.5|38.6|
> ||KVComm|**64.7**|**68.6**|**68.0**|
> |GSM8K|Original|81.1|**82.4**|**82.1**|
> ||DroidSpeak|52.2|20.6|14.5|
> ||KVComm|**81.5**|81.7|80.6|
> |HumanEval|Original|**86.3**|**83.9**|**84.5**|
> ||DroidSpeak|14.9|27.3|15.5|
> ||KVComm|81.4|83.2|83.2|
> |Math500|Original|**41.6**|**39.6**|42.6|
> ||DroidSpeak|14.8|14.8|14.8|
> ||KVComm|38.0|38.6|**44.2**|
>
> ## Q4: Performance on non-RoPE Models
> As discussed in Proposition 1, KVComm’s effectiveness relies on the regular positional structure provided by RoPE-based encodings. To validate this, we tested KVComm on OPT-6.7B (absolute position encoding). As shown in Table 4, the reuse rate drops to near zero.
>
> Table 4. Performance of OPT-6.7B on the MMLU benchmark.
>
> |Method|Accuracy|Reuse Rate|
> |-|-|-|
> |Original @ 2 Agents|25.5|-|
> |KVComm @ 2 Agents|24.8|0.6%|
> |Original @ 3 Agents|26.8|-|
> |KVComm @ 3 Agents|24.2|0.2%|
> |Orignal @ 4 Agents|25.5|-|
> |KVComm @ 4 Agents|22.2|0.16%|
>
> This is because **absolute position encodings do not preserve local positional relationships when the prefix changes, undermining cache similarity**. We will clarify in our revised manuscript that KVComm is designed specifically for RoPE-based models, and leave adaptation to other positional encodings as important future work.
>
>
> ## Q5: Discussion about KVComm in Dynamic Agentic Framework
>
> Thank the reviewer for highlighting the potential extension of KVComm. While our current implementation assumes predefined placeholders and templates to facilitate alignment and offset estimation, the core idea behind KVComm—context-aware offset estimation based on embedding similarity and anchor pools—is not inherently limited to static scenarios.
>
> In principle, **it can be applied wherever context change patterns recur, provided shared text segmentation is feasible** (as with techniques similar to automatic prefix caching in vLLM). Although our current work does not cover fully dynamic, unstructured cases, we recognize this as a meaningful direction and plan to extend KVComm for more dynamic agentic and debate-style benchmarks in future work.
>
> ## Q6: Reuse Rate Performance under Long-context Scenario
>
> We have provided the experimental results in Table 2 under the 4K context setting. It can be observed that when the cumulative positional deviations amplify, the overall reuse rate drops by 20%, from $\sim$ 70% to 50%. This aligns with expectations and the trend is similar to that of increasing placeholder number in the template, where the cumulative positional deviations will degrade the embedding similarity.

---

### Official Review · Reviewer_2PaM · 2025-07-03

**Clarity:** 2
**Significance:** 3
**Originality:** 2
**Rating:** 4
**Confidence:** 2

**Summary:**

The authors propose a method for reusing (prefilling) cache in multi-agent systems called KVComm.  The method is similar to CacheBlend, but is more specific to the multi-agent setting and uses 'anchors' dictated by changes in the custom prefixes/prompts for the different agents to inform where the KV cache should be shared.  Results on several benchmarks can show some degradation at large prefill rates, but the speedup for the method is large and the benefits increase with the number of agents.

**Questions:**

1. Could you be more specific in your claims on novelty?
2. Could you clarify the domain you're claiming your method should work well in? (specifically, do you intend for it to only be used in single-GPU, same-architecture situations?)
3. As someone not very familiar with the field, I'd love more intuition behind the anchors of your method and the closest related thing in multi-agent or other areas in the literature (response editing, speculative decoding)
4. It would be great to see experiments that line up with CacheBlend claims, and clear explanations for why that method is failing so badly on HumanEval / other times that it does.  Is the method dynamically recomputing the cache at a ~15% rate? Is this the vllm version?
5. Please bold the 'best' results rather than your results in Table 1 (e.g., CacheBlend is better on MMLU with 2 agents, but it's not obvious from the current table formatting)

**Ethical Concerns:**

["NO or VERY MINOR ethics concerns only"]

**Final Justification:**

The paper presents a method that addresses the important setting of efficient inference in multi-agent settings.  The authors clarified many points around limitations and differentiated novelty I had not seen clearly stated in the first edition of the paper.  Given that most of my concerns were around these areas, I have bumped my score by 1, and with sufficient clarity in the final version would advocate for an accept.

**Limitations:**

The authors should limit the scope of their claims to single-GPU, same-architecture settings very clearly and acknowledge possible hardware communication bottleneck impact on performance in different settings, since this drives a lot of other KV cache works' considerations.

**Paper Formatting Concerns:**

Two insufficient source citations:
[1] The proof and measurement of association between two things. 1961.
[23] Significant Gravitas. AutoGPT.

**Quality:**

2

**Strengths And Weaknesses:**

## Strengths
* The problem itself is an important one - how do you save computation when sending many of the same tokens through agents of the same model?
* The motivation for the method and general intuition is clearly presented
* There are a lot of interesting results / graphs presented.  If I knew the subfield better, I'd love to dig into those at a detailed level

## Weaknesses / Concerns
* Your positioning of KVComm within existing methods wasn’t clear to me:
  * You claim: “the first training-free, prompt-adaptive cache-sharing framework for efficient prefilling of multi-agent systems” <- why doesn't CacheBlend fit this?  Could you be more clear in your claim, since methods more general than multi-agent systems exist to do this? (The claim may be that it is more efficient specifically within multi-agent systems by leveraging X"
  * Is the method leveraging priors around prompts in a special way?
  * What does “context-aware” mean?
  * You mention and emphasize the assumption that other methods make that every agent runs the same model architecture.  It seems your method needs this assumption as well – is there a reason you emphasized it?  Could you make it clear that your method is similarly constrained (and taking advantage of this structure)
  * Given the drastic difference between your method and CacheBlend on HumanEval, I think you should (1) very clearly explain what this difference is due to, and (2) make it part of your major differentiator/impact (what other problems is this likely to have a big impact on?  Where is CacheBlend being used that it doesn’t perform well?)
  * Building up an anchor pool feels very similar to building up a database of past tokens themselves, i.e., speculative decoding methods – could you compare to them, at least in description / scenario?

* I’m concerned that the results don’t line up with the CacheBlend results and know the authors said they didn’t optimize the prompts for the open source models.  I think it’s important to rerun these experiments to match CacheBlend as best you can so the numbers are comparable between papers.

* Hardware/communications are usually a bottleneck in these situations.  It seems you don't address it in your paper - could you add clarification on:
  * The fact that this method is constrained to single-GPU settings (as far as I understand it)? (and homogenous networks, which is a common assumption)
  * Whether you needed to do anything special to store and retrieve the cache in an efficient way that was specific to this method?

## Smaller suggestions
* Cite “prior research” in “2 KV-Cache Sharing Scenario. Prior research has identified…”.
* In the same section, point out that your method is ‘Selective Recomputation’ (It’s kind of weird to make that #2 out of 4 – put it at the beginning or the end?  Or explain why you’re working on that one in particular in this setting and make it into a separate paragraph, perhaps?
* `Tab.` for table is hard to read.  Preference for `Table`

---

> ### Author Rebuttal · Authors · 2025-07-31
>
> Thank the reviewer for taking the time to review and providing the constructive and valuable feedback. Below, we present our point-by-point responses to each comment.
>
> ## Q1: More Clarification of KVComm's Contribution
>
> KVComm’s key contributions are threefold:
> - **Problem Formulation (Novel Motivation).** Prior KV-cache reuse methods (e.g., CacheBlend) target **single-agent** settings such as RAG, where multiple retrieved chunks are concatenated into one prompt and KV-caches are precomputed without external context. In **multi-agent systems**, however, each agent’s response is repeatedly consumed by successor agents. **The resulting KV-cache unavoidably encodes its original prefix context** due to autoregressive attention. As evidenced in our work, a naive adoption of previous methods in this case can fail in filtering the previous contextual information in the KV-cache, resulting in performance degradation. **We identify and address this multi-context redundancy as a distinct challenge for efficient prefilling in the multi-agent scenario, which to our knowledge, has not been covered by prior work**.
>
> - **Empirical Insight (Context-aware KV-cache Offset Proximity).** We observe that **for two similar tokens whose KV-caches are computed under the same prefix, when the prefix changes, their KV-cache deviations remain similar**. Unlike RAG-focused findings that KV-caches are largely prefix-robust, our measurements show this robustness breaks down in general multi-context (multi-agent) scenarios (see Figure 1 and Table 1 in the main text).
>
> - **New Reuse Paradigm (Prompt-Adaptive).** Instead of token-/layer-level selective recomputation (e.g., DroidSpeak, CacheBlend), KVComm performs **prompt-adaptive selection at the segment level**: a shared text is either fully reused or fully recomputed, based on its embedding similarity to stored anchors. We further design an **efficient KV-cache storage/retrieval system** that supports fast anchor lookup and long-context multi-agent pipelines.
>
> Therefore, based on the above, **KVComm is, to our knowledge, the first training-free, prompt-adaptive cache-sharing framework for efficient prefilling in multi-agent systems.**
>
> ## Q2: Priors Leveraged by KVComm
>
> The priors around prompts we use are consistent to the baseline multi-agent framework design. That is, KVComm follows the **baseline multi-agent dependency graph** to define each agent’s placeholder set and prefix segment set. IDs (for placeholders/prefix segments) are derived directly from the dependency graph. Details are in Appendix 6.2.2 and 6.3.2 of the supplementary material.
>
> ## Q3: Definition of Context-Aware Offset
>
> In Figure 2, the **context-aware offset is a KV-cache deviation induced by the (changed) prefix context**. When the prefix changes, the required offset to align a cached segment’s KV-cache with the new context changes accordingly, hence "context-aware".
>
> ## Q4: Domains / Assumptions for KVComm
>
> KV-cache reuse requires **identical model architecture** (tokenizer, computation graph, hidden size, #heads, #layers, etc.); otherwise KV-cache shapes/token spaces mismatch and reuse becomes infeasible without adaptation. KVComm also assumes **RoPE position encoding**, a common choice in modern LLMs, enabling explicit per-layer positional alignment across varying context lengths.
>
> ## Q5: Explanation of Performance Gap between KVComm and CacheBlend on HumanEval
>
> HumanEval agents produce code with many syntax separators (e.g., `. , ; !`), which induce **diverse and prefix-sensitive KV-cache distributions**. With CacheBlend’s 20% recomputation, many sensitive tokens remain stale, degrading successor agents’ understanding. KVComm instead globally applies the best anchor-based offset to approximate these tokens’ KV-cache under the new context. When we increase CacheBlend’s recomputation ratio to 80%, as shown in the following table, performance significantly increases but remains inferior to KVComm (>81% pass@1):
>
> |Reuse Rate|#Agent=2|#Agent=3|#Agent=4|
> |:-:|:-:|:-:|:-:|
> |20%|72.05|68.94|67.70|
> |80%|31.06|21.12|30.43|
>
> This suggests **coding tasks require near-full KV-cache updates** under context shifts, aligning with KVComm’s prompt-adaptive strategy.
>
> ## Q6: Comparison with Speculative Decoding
>
> At a high level, building up the anchor pool in KVComm might resemble building up a database of past tokens, such as those employed in speculative decoding methods. However, the fundamental concepts and underlying scenarios differ notably between KVComm and speculative decoding methods. Specifically,
>
> - **KVComm** accelerates **prefilling** across **multiple agents** by **reusing/aligning existing KV-caches** via anchor-based offsets. Each anchor in KVComm is essentially a cached representation of previously encountered segments along with their measured offsets from different prefix contexts. This allows us to approximate and reuse KV-caches dynamically at runtime.
> - **Speculative decoding** accelerates **decoding** in typically **single-agent** generation by **proposing and validating future tokens**. In speculative decoding, the model speculatively predicts based on previously generated tokens or precomputed continuations. If the prediction aligns closely with the actual model's output tokens, it accelerates decoding by accepting the speculative output. Otherwise, it discards the speculative output and recomputes fully.
>
> ## Q7: Fairness of Comparison with CacheBlend
> After carefully double examining the CacheBlend paper and our work, we would like to clarify the fundamental differences in application scenario and the faithfulness to CacheBlend in our implementation, which justify that **the experimental comparisonn with CacheBlend is as fair and objective as best we can**.
>
> **Scenario Difference:**
> - CacheBlend is designed/evaluated for **single-agent RAG**, reusing **non-prefix** chunks with small token-level recomputation. Consequently, the experiments reported in CacheBlend are specially designed for the RAG task in a single-agent setting, **which have no experimental results for multi-agent systems as we did.**
> - Our setting is **multi-agent**, where responses propagate across agents and prefixes shift frequently. **Our scenario involves general coordination among multiple agents rather than single-agent RAG contexts.**
>
> **Faithful Implementation of CacheBlend:**
> - We implemented CacheBlend per its paper and official code. To keep the similar reusing rate, we increase the recomputation rate to 20%, which theoretically further improves the performance of CacheBlend.
>
> **Comparability of Experimental Results:**
> - Our experiments intentionally use prompts representative of multi-agent coordination scenarios. Optimizing prompts specifically to match CacheBlend’s single-agent RAG scenario would not only distort the intended application scenario of our approach but also lose the realistic aspects and complexity of multi-agent inference.
> - As reported in Table 1 of the main text, CacheBlend performs reasonably on **RAG-like** tasks (e.g., MMLU with RAG agents; see Appendix 6.3.2), which align well with the claim of CacheBlend, but struggles on **math/coding** due to the reason analyzed in **Q5**. **Therefore, we have conducted as fair comparsion with CacheBlend as best we can.**
>
>
> ## Q8: Hardware / Communication Efficiency
>
> KVComm is **not** restricted to single-GPU. The two assumptions (identical architecture; RoPE) are **topology-agnostic**. We currently report single-GPU results due to framework API limitations that standard open-source inference frameworks (e.g., HuggingFace Transformers, vLLM, SGLang) focus mainly on prefix-based caching and lack stable public APIs for fine-grained, per-layer KV-cache injection across devices.
>
> **Solution to Distributed Communication Bottleneck:**
>
> Regarding the potential distributed communication bottleneck, KVComm can deliberately avoid moving entire KV-caches across GPUs/nodes. Instead, **we can synchronize only small-sized metadata (anchor indices and offset statistics) and perform KV-cache updates locally, which would remarkably reduce communication overhead**.
>
> We are currently integrating KVComm into distributed serving frameworks like vLLM and SGLang by adding minimal KV-cache read/write hooks to fully support efficient multi-agent systems in multi-GPU and distributed environments.
>
> **Cache Management Strategy:**
>
> Regarding the cache management strategy in KVComm, we designed **two three-level cache managers** to achieve efficient writing and retrieving anchors' KV-caches and the current shared KV-caches, respectively.
>
> |Anchor Manager|1st level|2nd level|3rd level|
> |-|-|-|-|
> |Indices|placeholder id, e.g., `user_question`, |anchor index, e.g., anchor[0]|agent id / embedding, e.g., `agent_1_ph_delta`|
> |Values|Anchor list|dict of different KV-cache offset in different agents and the anchor embedding tensor|KV-cache/embedding tensor|
>
> |Shared KV-cache Manager|1st level|2nd level|3rd level|
> |-|-|-|-|
> |Indices|agent id / user_input, e.g., `agent_1`|placeholder_id, e.g., `response`|turn index, e.g., response[-1]|
> |Values|dict of different agents' response KV-caches and outside input KV-caches|dict of response and prefix KV-caches|KV-cache list|
>
> Based on these two cache managers, each agent can quickly retrieve their intended KV-caches and also store their generated KV-caches. Meanwhile, we conduct the process of KV-cache storage and retrieval in an asynchronous manner, thus further improving the efficiency. For clearer understanding, we will opensource the code repository once accepted.
>
> ## Q9: Paper Structure & Formatting
>
> We will revise Section 2.2 to clarify positioning among KV-acceleration methods, adjust `Table` formats to avoid misinterpretation, and verify all references' formats.

---

> > ### Author Response · Authors · 2025-08-06
> >
> > Dear Reviewer,
> >
> > We sincerely appreciate your valuable time and detailed feedback on our manuscript. Your insightful comments have greatly contributed to improving the quality of our paper, particularly regarding:
> >
> > 1. Clarification of KVComm's novelty in terms of motivation, insights, and paradigms;
> >
> > 2. Detailed elaboration on the priors, assumptions, and limitations associated with KVComm;
> >
> > 3. Clear explanations of technical terminology within KVComm;
> >
> > 4. In-depth discussions comparing KVComm with advanced acceleration paradigms such as CacheBlend and speculative decoding;
> >
> > 5. Insightful analysis explaining KVComm’s significant advantages over CacheBlend in mathematical reasoning and coding tasks;
> >
> > 6. A thorough introduction of the memory management strategies specifically developed for efficient KV-cache operations across agents in KVComm;
> >
> > 7. Comprehensive discussions on deploying KVComm within distributed multi-agent scenarios;
> >
> > 8. Detailed revisions addressing citation and table formatting issues.
> >
> > We deeply value your contributions and insights, which have been instrumental in refining our work. As we approach the end of the discussion phase, we kindly wish to inquire whether our responses have addressed your concerns effectively. If any further clarification or discussion is needed, we remain eager and available to provide it.
> >
> > Thank you once again for your thoughtful and constructive feedback!
> >
> > Best regards,
> > Authors of Submission 1343

---

> > ### Comment · Reviewer_2PaM · 2025-08-07
> >
> > Thank you for the detailed rebuttal.  Each of these points greatly increased clarity for me, and in seeing that other reviewers wanted clarification on similar points, I'd like to make sure the authors plan to ensure these points are made clear in the final version of the paper, especially around limitations, assumptions, and differentiated novelty.
> >
> > Increasing my rating to match the increased clarity on the limitations and differentiated novelty.

---

> > > ### Author Response · Authors · 2025-08-07
> > >
> > > We greatly appreciate the reviewer's positive feedback and sincerely thank the reviewer for increasing the score. We will follow all reviewers' valuable suggestions to ensure all points are clarified in the final version of the manuscript, and open-source the code repository that integrates KVComm with modern LLM serving frameworks to facilitate the deployment of efficient multi-agent systems.

---

### Comment · Area_Chair_78cd · 2025-08-09

Hi all, thanks for the active discussion and engagement! All reviewers agree the work is in good shape, and the authors have addressed most of the concerns from the first review round.

All reviewers have added the Mandatory Acknowledgement, so everything is within reach. Please feel free to jump in for further discussions if any — less than 24 hours remain in the discussion period. Thank you!

---

### Decision · Program_Chairs · 2025-09-17

**Decision:**

Accept (poster)

**Comment:**

This submission proposes a training-free framework (KVComm) for efficient KV-cache reuse in multi-agent LLM systems. In multi-agent settings, overlapping contexts often lead to redundant KV-cache computation due to diverging prefixes. This would prevent straightforward cache reuse. KVComm introduces an online anchor pool that records cache deviations under varying prefixes and uses them to align offsets dynamically. This idea helps enable effective cache sharing across agents. Experiments on show performance improvements, including up to 13× speedup in prefill latency and over 70% KV-cache reuse.

Strengths of this submission:

1. Reviewers 2PaM, y5Xb consider that it tackles an important efficiency problem in multi-agent LLM systems. The proposed method addresses the overhead caused by repeated KV-cache recomputation in multi-agent pipelines.
2. reviewers TNJW, 3BgL are happy with its empirical results, as experiments show consistent acceleration (13× speedup) and thorough ablations and sensitivity analyses that validate the method’s design choices.
3. Reviewers y5Xb, 3BgL mentioned that the framework is supported by insights into KV-cache offset alignment while remaining training-free and applicable in real-world multi-agent scenarios.

Weaknesses of this work:

1. Though presented with promising performance gains, three reviewers (2PaM, TNJW, 3BgL) also point out that its comparison with existing methods does not form as its strong point. Specifically, while CacheBlend is discussed, direct experimental comparisons and positioning relative to related works such as speculative decoding and DroidSpeak are insufficient.
2. reviewers 2PaM, y5Xb also identify clarity and presentation issues. Some parts of the paper including figures and terminology are not easy to follow, making the implementation details of KV-cache offset alignment less intuitive for readers unfamiliar with the subfield.

In summary, the submission makes contributions to improving efficiency in multi-agent LLM systems and provides solid empirical results supported by insights. While there are concerns regarding baseline comparisons, applicability to heterogeneous settings, and some presentations issues, the technical soundness and performance improvements make this a valuable addition to the literature. During the rebuttal, the reviewers are generally in support of this work (all four final ratings are positive). All factors considered, this work is given an acceptance recommendation.